



# Reversible ice sheet thinning in the Amundsen Sea Embayment during the Late Holocene

Greg Balco[1,10], Nathan Brown[2,10], Keir Nichols[3,10], Ryan A. Venturelli[4,10], Jonathan Adams[3,6], Scott Braddock[5], Seth Campbell[5], Brent Goehring[7], Joanne S. Johnson[6], Dylan H. Rood[3], Klaus Wilcken[8], Brenda Hall[5], and John Woodward[9]

[1]Berkeley Geochronology Center, Berkeley CA USA
[2]University of Texas, Arlington, Arlington TX USA
[3]Imperial College London, London, UK
[4]Colorado School of Mines, Golden CO USA
[5]University of Maine, Orono ME USA
[6]British Antarctic Survey, Cambridge, UK
[7]Tulane University, New Orleans LA USA
[8]ANSTO, Lucas Heights NSW, Australia
[9]Northumbria University, Newcastle UK
[10]These authors contributed equally to this work. Listed in alphabetical order.

**Correspondence:** Greg Balco (balcs@bgc.org)

**Abstract.**

Cosmogenic-nuclide concentrations in subglacial bedrock cores show that the West Antarctic Ice Sheet (WAIS) at a site between Thwaites and Pope Glaciers was at least 35 m thinner than present in the past several thousand years, and subsequently thickened. This is important because of concern that present thinning and grounding line retreat at

5 these and nearby glaciers in the Amundsen Sea Embayment may be irreversible, potentially leading to decimeter- to meter-scale sea level rise within decades to centuries. A past episode of ice sheet thinning, which took place in a similar although not identical climate, was not irreversible. We propose that the past thinning-thickening cycle was due to a glacioisostatic rebound feedback, similar to that invoked as a possible stabilizing mechanism for current grounding line retreat, in which isostatic uplift caused by early Holocene thinning led to relative sea level fall favoring grounding line

advance.

## 1 Overview and background

During the 2019-20 Antarctic field season we used a lightweight minerals exploration drill adapted for polar use (Boeck-mann et al., 2021) to collect four bedrock cores from 30-40 m beneath the ice sheet surface on the northern side of the Mount Murphy massif, a volcanic edifice located between Thwaites and Pope Glaciers in the Amundsen Sea Embayment

of West Antarctica (Figs. 1, 2). These and nearby glaciers are experiencing rapid grounding line retreat and thinning, with the result that this sector of Antarctica is the largest contributor to global sea level rise at present (Shepherd et al., 2018; Smith et al., 2020). Retreat and thinning of major Amundsen Sea glaciers is of concern because these glaciers



are retreating into overdeepened basins in which ice is grounded well below sea level. Mercer (1978) and subsequently many others have proposed that this geometry is inherently unstable because, in general, the rate of ice loss at a marine ice margin increases with the water depth, so ice margin retreat into an overdeepening will result in a runaway increase in the retreat rate that is irreversible until the entire overdeepening is fully deglaciated. As the overdeepening in this case

underlies much of the present WAIS, this would imply that the present episode of ice thinning and grounding line retreat in this region will inevitably lead to loss of a significant portion of the WAIS, with globally significant sea level impacts. Thus, recent modeling studies (Favier et al., 2014; Joughin et al., 2014; Rosier et al., 2021) have focused on whether or not further retreat inboard of present grounding line positions implies irreversible deglaciation of large segments of West Antarctica.

The purpose of the drilling program was to apply luminescence dating and cosmogenic-nuclide exposure dating of subglacial bedrock to establish whether or not similar thinning episodes have taken place in the geologically recent past. These methods are complementary, but not equivalent. Luminescence dating is a means of detecting past exposure of mineral grains to sunlight (Aitken, 1998) that can be used to determine whether a rock surface that is ice-covered now was ice-free in the past. Detection of past ice sheet thinning using luminescence dating requires that a subglacial rock

surface was exposed to sunlight in the past, which can only occur if the surface is completely ice-free (or nearly so, depending on the optical properties of the basal ice). Cosmogenic-nuclide exposure-dating, on the other hand, relies on trace nuclides that are produced by cosmic-ray interactions in near-surface rocks and minerals, but are not produced in surfaces covered by more than several meters of ice (Dunai, 2010; Balco, 2011). The cosmic-ray flux is gradually, not instantaneously, attenuated with depth below the ice surface, so cosmogenic-nuclide measurements can be used to

detect past thinning of the ice above a subglacial bedrock surface even if the surface was never completely ice-free.

A much more common use of cosmogenic-nuclide exposure-dating is to estimate the deglaciation age of rock surfaces that are ice-free now and were exposed by past deglaciation. This approach has been widely applied to glacial deposits preserved above the present ice surface throughout Antarctica to reconstruct ice sheet change during times in the past when ice sheets were larger than they are now (e.g., Stone et al., 2003). Reconstructing ice sheet change during warm

periods when ice sheets were smaller, on the other hand, is necessary for predicting response to projected climate warming, but more difficult because the evidence that an ice sheet was smaller is inaccessible beneath the present ice sheet (Johnson et al., 2022). Here we show that this obstacle can be overcome by a targeted drilling program.

Exploiting past exposure of subglacial bedrock as a means of detecting ice sheet thinning is only possible at a site where bedrock is above sea level and thinning will progressively increase the extent of exposed rock adjacent to the

ice margin (Spector et al., 2018). The Mt. Murphy massif is the only such location in the vicinity of Thwaites and Pope Glaciers. On the north side of the massif, several ice-free ridges descend towards Pope Glacier and Crosson Ice Shelf (Figure 1). At one of these ridges, the north ridge of Kay Peak, a subsidiary peak of the Mt. Murphy massif (henceforth "Kay Peak Ridge"), we collected a series of bedrock samples on the exposed part of the ridge from the ice margin to 200 m above the ice surface, and four bedrock cores from beneath 30-40 m of ice on the subglacial extension of the ridge.





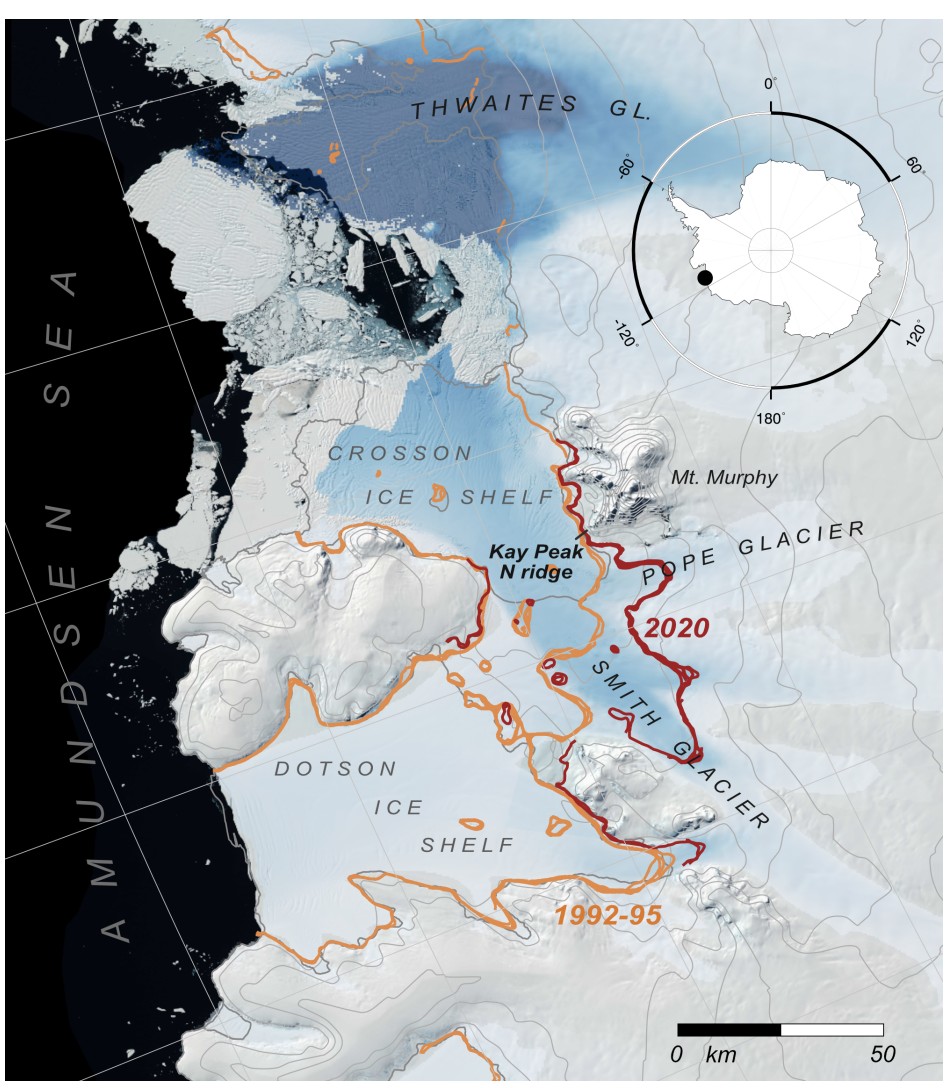

**Figure 1.** Map of central Amundsen Sea coast generated using data sets from Quantarctica version 3 (Matsuoka et al., 2021) and showing major glaciers, drill site, and historical grounding line positions in the Thwaites-Pope-Smith Glacier region. The 1992-95 and 2020 grounding line positions are from Rignot et al. (2011) and Milillo et al. (2022) respectively. Blue shading qualitatively shows areas of rapid ice flow and highlights major glaciers.



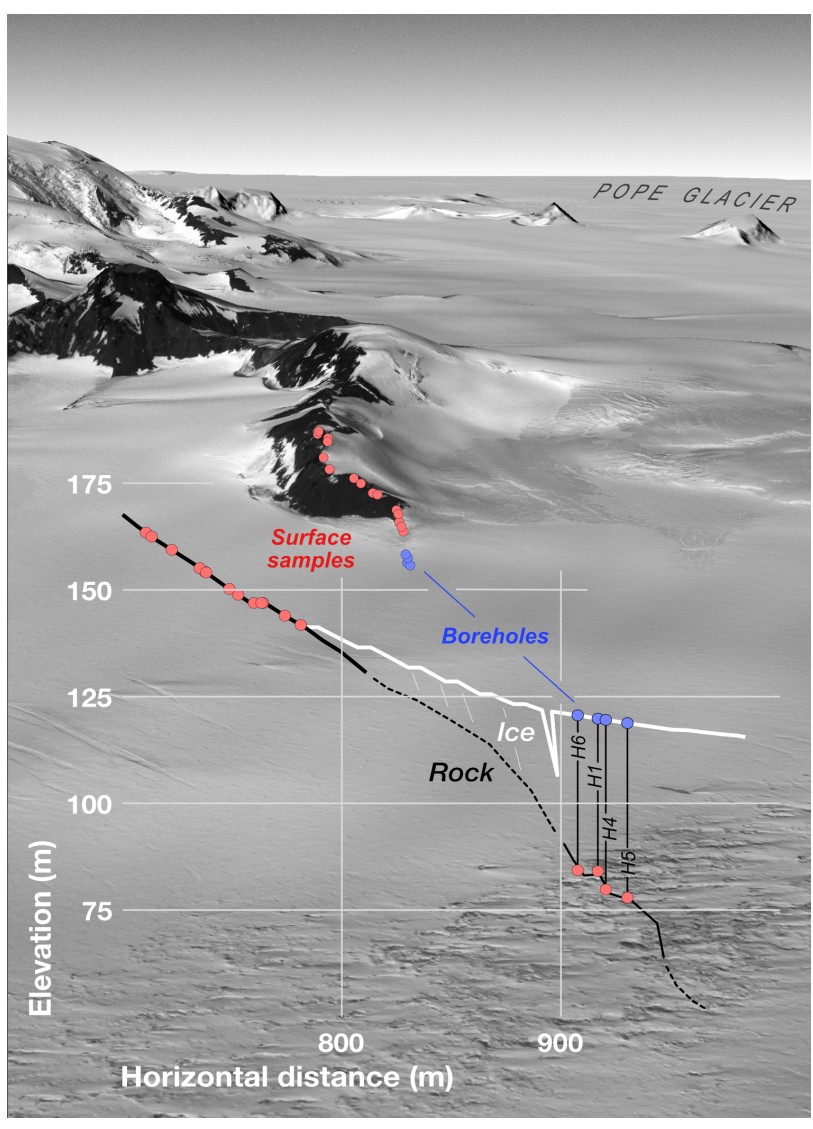

**Figure 2.** Synthetic oblique view of the drill site at the north ridge of Kay Peak generated from Worldview imagery (© 2018, Maxar) and the REMA DEM (Howat et al., 2019). View direction is to the S, as indicated in Fig. 1, from the position of a viewer located over the Crosson Ice Shelf and looking up Pope Glacier. The overlay is an elevation profile along the axis of Kay Peak Ridge showing the relationship between samples collected from the exposed portion of the ridge, drill sites, and subglacial bedrock samples.



It is likely that the ice thickness at Kay Peak Ridge is coupled to the position of the grounding line of Pope Glacier. Like Thwaites Glacier, Pope Glacier has experienced rapid grounding line retreat in recent decades, from immediately adjacent to our drill site in 1992-95 to ~10 km landward of the site in 2014 (Milillo et al., 2022) (Fig. 1). Correspondingly, comparison of 1966 aerial photographs with 2018-19 satellite imagery indicates expansion of the area of exposed rock on the ridge, which requires thinning of adjacent ice by at least 31 m in 52 years (Fig. 3). Lacking surface mass balance data for the 1966-2020 period, it is not possible to prove conclusively that ice thinning at the drill site is solely a dynamic response to nearby grounding line retreat, but the coincidence of thinning and retreat is consistent with this mechanism.

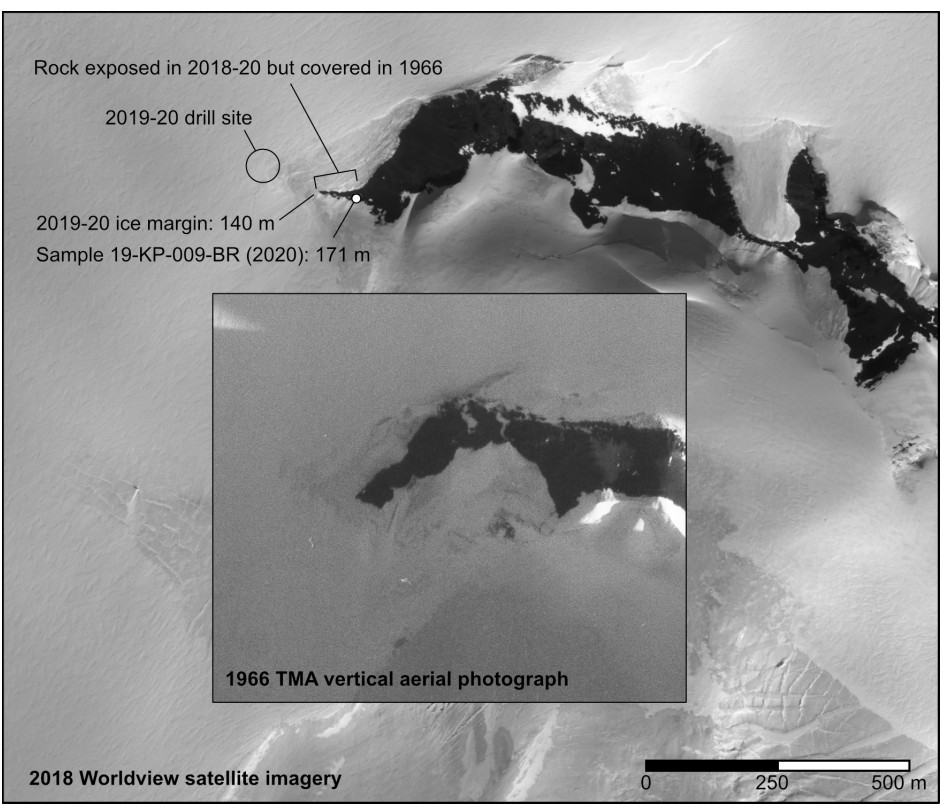

**Figure 3.** Worldview satellite imagery (© 2018, Maxar) collected December 8, 2018 (main image) showing N end of Kay Peak Ridge, compared with U.S. Navy vertical trimetrogon aerial (TMA) photograph (inset image) taken Jan 5, 1966 (TMA1720, frame 231V). The 1966 photograph is not geolocated, so the images were adjusted to a common scale by matching bedrock features on the lower ridge. This comparison does not attempt to correct for look angle, so comparison of the extent of exposed rock is not quantitative. However, the shape of the ice margin clearly indicates that a portion of the lower ridge exposed in 2018 was not exposed in 1966. In field work in 2019-20, we collected samples from the portion of the ridge that was ice-covered in 1966, and surveyed sample locations and the lowest exposed rock on the ridge using a combination of handheld GPS, differential GPS, and barometric elevation measurements. Based on the image comparison, the highest sample that was clearly ice-covered in 1966 has a present elevation of 171 m. As the present ice margin is at 140 m elevation, the elevation of the ice margin decreased at least 31 m between 1966 and 2020.





Exposure-age data from the ice-free portion of the ridge, collected in previous research (Johnson et al., 2020) and supplemented by additional measurements in this work (Supplementary Methods 2; Supplementary Tables S1, S5), include measurements of in-situ-produced $^{14}$C and $^{10}$Be in quartz (Fig. 4). The important difference between $^{14}$C and $^{10}$Be exposure ages is that the 5730-year half-life of $^{14}$C precludes the survival of any significant $^{14}$C concentration

from surface exposure prior to the Last Glacial Maximum (LGM) 15,000 - 25,000 years ago. Although measured $^{10}$Be concentrations could be, and often are, the result of surface exposure during both the present interglaciation and one or more previous ones, $^{14}$C can only be the result of post-LGM exposure (Nichols et al., 2019). As described by Johnson et al. (2020), $^{10}$Be exposure ages from bedrock and erratics on Kay Peak Ridge are spuriously old due to $^{10}$Be inheritance, but $^{14}$C exposure ages from the ridge agree with a large set of other data from the overall Mt. Murphy massif in recording

progressive exposure of the ridge by early Holocene ice sheet thinning from 200 m above the present ice surface at 6-10 ka to near present conditions at 4-7 ka. $^{10}$Be exposure ages from a nearby small outcrop 1 km to the west of Kay Peak Ridge (Fig. 4, Adams et al., 2022) also indicate deglaciation 6-7 ka. Although the $^{14}$C exposure ages from Kay Peak Ridge scatter more than expected from measurement uncertainties, likely because of the complex effect of both regional ice surface lowering and retreat of fringing icefields in uncovering the ridge axis, they agree with other exposure-age

data sets throughout the Amundsen Sea region that uniformly show rapid early and middle Holocene thinning reaching the present ice surface between 4-7 ka (Johnson et al., 2014, 2020). Early Holocene thinning was associated with, and presumably caused by, coeval retreat of glacier grounding lines across the adjacent continental shelf (Hillenbrand et al., 2017).

There are no exposure-age data in the Amundsen Sea region indicating thicker than present ice after 4 ka (Johnson

et al., 2022). Thus, existing observations require either (i) zero change in ice thickness in the last several thousand years, or (ii) continued thinning below the present ice surface, followed by thickening to the present configuration. Stable ice thickness for millennia appears unlikely given dynamic late Holocene boundary conditions, including relative sea level (RSL) change forced by eustatic and glacioisostatic effects, climate and oceanographic changes (Walker and Holland, 2007; Hillenbrand et al., 2017), and changes in grounding line position elsewhere in Antarctica (Venturelli et al., 2020;

King et al., 2022). In addition, apparent $^{14}$C exposure ages of 4-7 ka on samples below the 1966 ice surface (Fig. 4) require at least some period in the Holocene during which the ice was thinner than it was in 1966.

Relative sea level in coastal Antarctica rose between the LGM and early Holocene because meltwater addition to the ocean from disintegrating LGM ice sheets outpaced local isostatic rebound, but this relationship reversed in the late Holocene as global ice volume stabilized but isostatic equilibration to early Holocene mass loss was not yet complete

(Lambeck et al., 2014). As relative sea level is a primary control on grounding line position at marine ice margins (e.g., Gomez et al., 2010), this sequence of events predicts grounding line advance and associated thickening in the late Holocene (Johnson et al., 2022), and isostatic response to ice sheet thinning has also been proposed as a potential stabilizing feedback for present grounding line retreat and thinning (e.g., Barletta et al., 2018). The aim of the drilling program was to determine if this process was active in the past by detecting and, if present, quantifying, a hypothesized

Holocene thinning-thickening cycle.

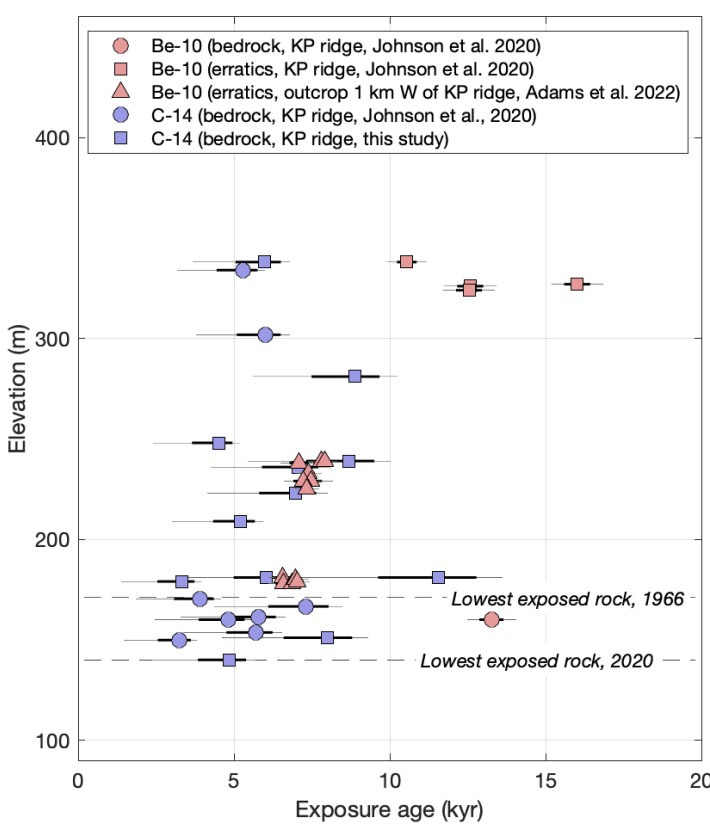

**Figure 4.** Apparent $^{10}$Be and $^{14}$C exposure ages for bedrock and erratic samples from the ice-free section of Kay Peak Ridge, including data already published in Johnson et al. (2020) and additional measurements in this study (see Supplementary Methods 2 and Supplementary Tables S1 and S5. All data are also available through the 'ICE-D:ANTARCTICA' database at http://www.ice-d.org). Apparent $^{10}$Be exposure ages from an adjacent outcrop 1 km west of Kay Peak Ridge, described in Adams et al. (2022), are also shown. Error bars indicate 68% (thick line) and 95% (thin line) confidence intervals. 'Apparent exposure ages' are exposure ages calculated from measured nuclide concentrations with the assumption that samples have experienced a single period of exposure at their present location without burial or surface erosion.



## 2   Drilling and sample collection

We located the subglacial extension of Kay Peak Ridge using ice-penetrating radar. Ice within ∼200 m of the lowest exposed rock on the ridge was crevassed and unsafe for drilling, so the survey began at the closest accessible point to the exposed ridge, where ice was ∼25 m thick, and extended away from the ice margin until the ridge crest was below the
depth feasible for drilling (∼ 60 m). This identified an approximately 70-m-long section of subglacial ridge where drilling was possible (Figs. 2, 5).

We conducted the survey with a ski-towed radar system consisting of a Geophysical Survey Systems, Inc. (GSSI) SIR-4000 control unit and model 3108 100 MHz monostatic transceiver, coupled to a Garmin GPSMap 78 handheld GPS unit. We also installed a series of snow stakes in a 70 x 100 m grid with 10 m spacing overlying the subglacial
ridge, and marked these stakes on radar profiles collected along grid lines (Fig. 5). This provided higher-resolution spatial referencing than possible using the GPS alone. The antenna was oriented perpendicular to the direction of travel, and towed at approximately 1 km h$^{-1}$. Scans were recorded for 2400 ns with 1024 32-bit samples per scan and 30-50 scans per second, translating to ∼ 90 scans per meter. Data were recorded with range gain and post-processed with (i) high (25 MHz) and low (300 MHz) pass filters to reduce noise; (ii) distance corrections based on the 10-meter-spaced grid
stakes; and (iii) stacking (5-10 stacks) to improve the signal-to-noise ratio. We then applied a basic Kirchoff migration of the data using hyperbolae from bedrock and internal reflectors to estimate relative permittivity and associated velocities and depths to bedrock. We did not apply more advanced processing techniques in the field because the survey was conducted immediately (1 day) before we began drilling. Hence, we chose drill sites as close as possible to the ridge axis by locating the apparent high point on each ridge crossing in the migrated data.

We collected subglacial bedrock samples using a two-stage drilling process. First, we made an access hole through the ice, extending as close as possible to the bedrock surface, using a Badger-Eclipse cable-suspended ice core drill, which produces a 113 mm borehole. Drilling with this system stopped when the sediment concentration in basal ice became high enough to dull the cutting bit and halt progress. We then occupied the access hole with a "Winkie" pipe-string, normal circulation compact rock coring drill, modified for polar use by the U.S. Ice Drilling Program (Boeckmann et al., 2021).
We set casing in the ice at the bottom of the access hole using an inflatable packer to enable fluid circulation, and cored through the remaining sediment-rich basal ice into underlying bedrock.

We reached bedrock 36-41 m below the surface at four boreholes (Fig. 2, Supplementary Table S1). In all boreholes, we observed a firn-ice transition 12-17 m below the surface, and then a transition from clean glacier ice to a 40-120 cm layer of sediment-rich basal ice in contact with bedrock. Note that the presence of debris-rich, optically opaque basal
ice is important in this study because resetting of luminescence signals could not occur through thin ice; detection of ice thinning using luminescence dating would require complete removal of basal ice and direct sunlight exposure of the bedrock surface. At three holes, we recovered 114- to 137-cm-long, 33 mm diameter cores of the underlying bedrock, whose lithology is a biotite gneiss identical to the bedrock exposed at the ice margin. At the fourth borehole (19-KP-H6) we recovered 7 cm of fragmented rock with the same lithology, but were not able to determine whether it was part





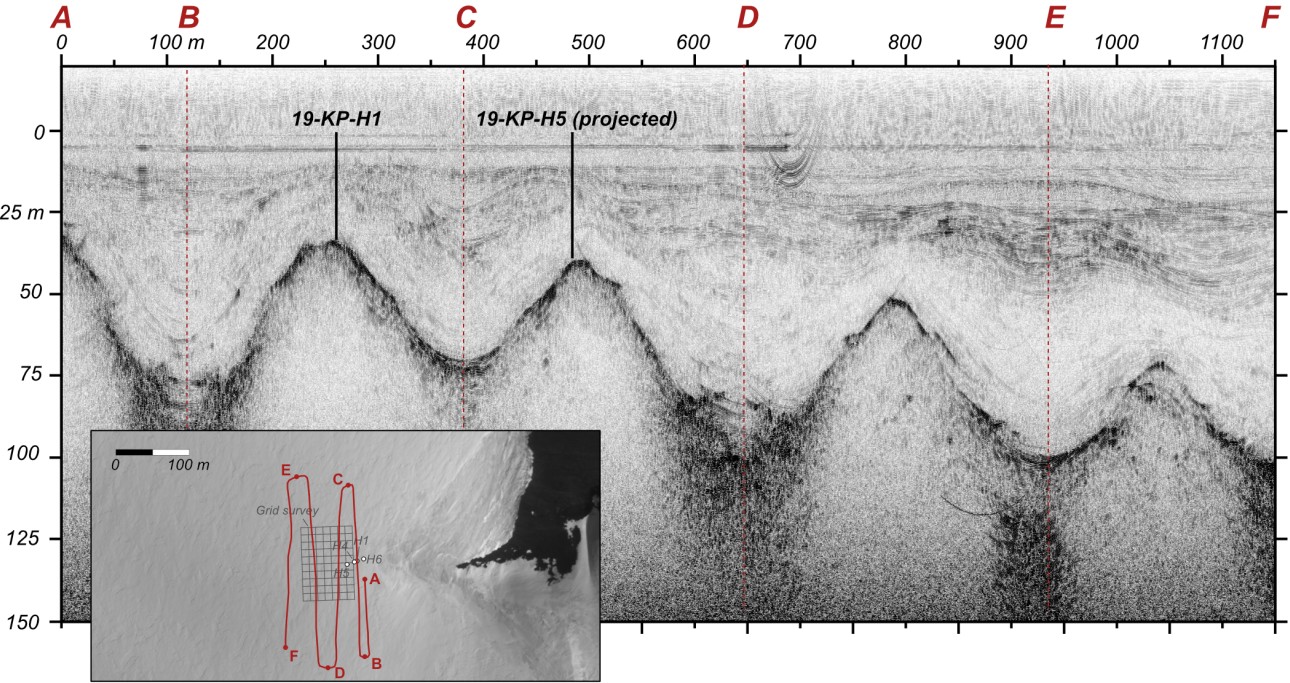

**Figure 5.** 100 MHz radar profile collected prior to drilling in 2019-20 that crosses the subglacial portion of Kay Peak Ridge four times (see inset map). The profile has been distance normalized and a Kirchoff migration was applied to improve bedrock depth estimates. Core 19-KP-H1 lies on the radar line, as shown, and the nearest projected location of 19-KP-H5 is also shown. The projected locations of two other cores, 19-KP-H4 and 19-KP-H6, would be near 19-KP-H1 in this figure. The location of the grid survey that was also used for drill site selection is shown in gray.

of the bedrock surface or a loose clast within the basal ice. At two of the boreholes, we were able to extract bedrock core tops from the core collection barrel within a lightproof bag so that they were not exposed to sunlight, permitting infrared stimulated luminescence (IRSL) measurements on the bedrock surface. In the next sections, we describe IRSL measurements on two core tops and measurements of the cosmic-ray-produced radionuclides $^{10}$Be, $^{14}$C, and $^{26}$Al in quartz from three cores.

## 3   Infrared stimulated luminescence (IRSL) depth profiles

### 3.1   Methods

We made IRSL depth profile measurements (Sohbati et al., 2011, 2012) in the uppermost 4 cm of the two core top samples that we were able to remove from the core barrel without exposure to sunlight (19-KP-H1, 19-KP-H4). In addition,





to establish kinetics of bleaching during sunlight exposure for this lithology, we collected equivalent depth-profile data on a rock surface above the present ice margin that had been newly exposed by collection of a rock sample four years earlier (Johnson et al., 2020).

Depth-profile measurements were made by slicing rock samples parallel to the exposed surface using a low-speed, water-cooled lapidary saw with a 0.38-mm fine-kerf blade. Each slice was then approximately broken into quarters, each of which was disaggregated using gentle crushing in a mortar and pestle to yield mineral grains suitable for IRSL measurement, and all were analyzed separately in order to better capture the natural variability in luminescence response at each depth. We then used a post-infrared infrared-stimulated luminescence (pIRIR) protocol (Buylaert et al., 2009) to determine sensitivity corrected natural luminescence intensities ($L_N/T_N$) as well as the dose response and fading characteristics of each sample. This pIRIR protocol yields measurements of two luminescence signals: the more bleachable $IR_{50}$ signal, which originates from both Na- and K-rich feldspars and has higher signal fading rates, and the less bleachable $pIRIR_{225}$ signal, which is mostly from K-rich feldspars and fades less (Thomsen et al., 2018). Further details of the measurements are in Supplementary Methods 1 and Supplementary Table S2.

### 3.2 Results

The depth profile in the surface sample shows significant bleaching, manifested as values of $L_N/T_N$ for both luminescence signals that approach zero at the surface and are significantly below field saturation values in the uppermost 1 cm, after 4 years of exposure (Fig. 6). If the core tops had been exposed to sunlight during the Holocene, they would also show values of $L_N/T_N$ systematically below saturation at shallow depths. They do not. Rather, IRSL signals in core tops are indistinguishable from calculated field saturation at all depths. This means that the bedrock surfaces of the core tops were not exposed to sunlight during the period required to approach saturation. We estimate the period required to approach saturation as 200 - 280 ka based on estimated dose rates and measured fading properties (Supplementary Methods 1). Therefore, bedrock at these two core sites was not ice-free during the Holocene.

### 4 Cosmogenic-nuclide measurements

### 4.1 Methods

We sectioned bedrock cores into 10-cm segments, separated quartz from each segment, and measured concentrations of cosmic-ray-produced [10]Be and [14]C in quartz separates from three cores, and additionally [26]Al in quartz from one core. As discussed above in the introduction, the purpose of measuring multiple cosmogenic nuclides in the core samples is to distinguish Holocene from pre-Holocene exposure of bedrock. [10]Be and [26]Al have relatively long half-lives (1.4 Ma and 0.7 Ma, respectively), so if the core sites were not affected by subglacial erosion during the LGM, [26]Al and [10]Be produced in near-surface rocks in a previous ice-free period prior to the LGM could be preserved to the present. Observing cosmogenic [10]Be or [26]Al in subglacial bedrock would prove that the surface had been exposed at some time in the past, but would not prove that the surface had been exposed in the Holocene. In constrast, [14]C has a half-life of





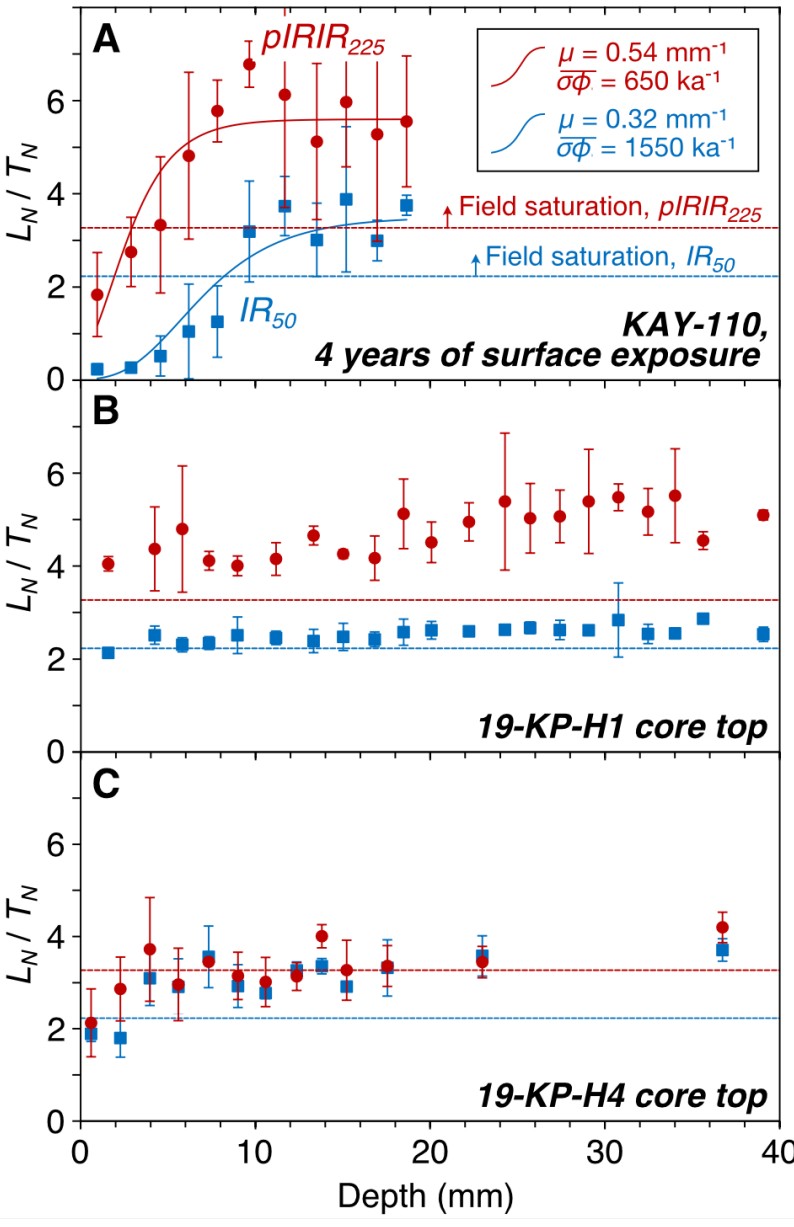

**Figure 6.** Sensitivity-corrected luminescence intensity for $IR_{50}$ and $pIRIR_{225}$ signals, the $IR_{50}$ signal being more bleachable when exposed to sunlight. The depth profiles for both signals in both core tops (**B,C**) are indistinguishable from field saturation, the 95% confidence level for which is indicated with horizontal dashed lines. By comparison, KAY-110 (**A**), which was exposed to sunlight for four years prior to measurement, shows reduced intensity near the surface due to signal bleaching. For KAY-110, solid lines show a bleaching model fitted to the data to yield estimates of the detrapping rate at the rock surface, $\overline{\sigma\phi}$ (ka$^{-1}$) and the sunlight attenuation coefficient, $\mu$ (mm$^{-1}$) (Supplementary Methods I; Supplementary Table S2).





5730 years, so $^{14}$C produced prior to the LGM would be removed by radioactive decay. Cosmogenic $^{14}$C observed in subglacial bedrock unambiguously requires Holocene exposure.

In brief, $^{26}$Al and $^{10}$Be analysis employs an isotope dilution method in which quartz is dissolved in hydrofluoric acid in the presence of a known quantity of the common isotopes $^9$Be and $^{27}$Al, Be and Al are separated by column chromatography, and $^{10}$Be and $^{26}$Al concentrations are quantified by accelerator mass spectrometer (AMS) measurement of $^{10}$Be/$^9$Be and $^{26}$Al/$^{27}$Al ratios. In-situ-produced $^{14}$C in quartz is measured by melting of the quartz under vacuum to release $^{14}$C as $CO_2$, followed by $CO_2$ purification, barometric measurement of the total amount of $CO_2$, and conversion to graphite for AMS measurement of the $^{14}$C/$^{12}$C ratio. The detection limit for all three nuclides is quantified by measurement of process blanks. $^{10}$Be and $^{26}$Al concentrations in core samples are 2000-7000 and 16000-44000 atoms g$^{-1}$, 2-5 times the detection limit inferred from process blanks and within the range routinely measured for exposure-dating applications. $^{14}$C concentrations in quartz in the core samples are in the range 10,000 - 50,000 atoms g$^{-1}$. These concentrations are 2-3 times typical detection limits for in-situ $^{14}$C in quartz and are significantly lower than routinely measured. Thus, we characterized the distribution of process blanks by analyzing 50 process blanks during the period in which sample analysis took place. This showed that the analytical background is not well approximated by a normal distribution, and allowed an improved background correction that takes account of the observed, positively skewed distribution. In addition, we made replicate analyses of multiple aliquots of quartz for nearly all samples. Analytical methods and blank correction procedures are are described in detail in Supplementary Methods 2 and SupplementaryTables S3-S5.

## 4.2 Results

$^{10}$Be and $^{26}$Al concentrations decrease with depth in all cores (Figure 7). Thus, $^{10}$Be and $^{26}$Al concentrations are the result of near-surface cosmic-ray production. Although $^{10}$Be and $^{26}$Al concentrations in one core conform to the production ratio and are therefore consistent with recent exposure, the presence of cosmic-ray-produced $^{10}$Be and $^{26}$Al alone, as discussed above, does not necessarily show that ice at this site was thinner during the Holocene. Observed concentrations could have been produced during an interglacial period prior to the LGM and not subsequently removed by subglacial erosion.

Although $^{14}$C concentrations in quartz from cores are, as noted above, lower than routinely measured and near detection limits, concentrations of $^{14}$C that are unequivocally above background are present in core samples. As for $^{10}$Be and $^{26}$Al, $^{14}$C concentrations are inversely correlated with depth in all cores (Fig. 7), again requiring near-surface cosmic-ray production. The $^{10}$Be-$^{26}$Al data require cosmic-ray exposure at some point, not necessarily in the Holocene, but the $^{14}$C data require Holocene cosmic-ray exposure.





**Figure 7.** [10]Be and [14]C concentrations in core samples compared to the null hypothesis for zero Holocene thinning and to results of the random search algorithm. Columns **A** and **B** compare measured and predicted in-situ [10]Be and [14]C concentrations in quartz in three subglacial bedrock cores. Green and gray lines show concentrations predicted by ice thickness change histories generated by the random search algorithm that best fit data from each core. Results shown in green are the best-fitting 1% of the random search iterations, and others are shown in gray. Likewise, column **C** compares the thickness change histories that do and do not fit the core data to [10]Be and [14]C exposure-age data from above the present ice surface on the north ridge of Kay Peak. The same exposure-age data are redundantly shown in all three panels in column **C**. [26]Al data for core 19-KP-H1 are equivalent to [10]Be results and are omitted here for clarity (Supplementary Methods 3). In all plots, 68% and 95% confidence intervals on nuclide concentrations and exposure ages are shown as thick and thin error bars. Black lines labeled 'Null hypotheses' represent a range of variations on a null hypothesis scenario in which the ice sheet thins rapidly to its present thickness in the early to middle Holocene and does not change thereafter. Predicted nuclide concentrations for the null hypothesis are also shown as black lines in the left and middle columns (very close to the y-axes), highlighting that they are not consistent with observed concentrations.

In contrast to the IRSL data, therefore, the cosmogenic-nuclide data require that ice at the core sites has been thinner sometime during the Holocene than it is now. We show this quantitatively by considering a null hypothesis that (i) any preexisting nuclide inheritance was removed by subglacial erosion during the LGM; (ii) the ice thinned rapidly during the early to middle Holocene, as indicated by exposure-age data from ice-free areas in the region; (iii) the ice thickness
reached its present value between 4-7 ka, and (iv) the ice thickness did not change subsequently. In this hypothesis, ice overlying core sites was never thinner than present, so [10]Be, [26]Al, and [14]C production rates were never higher than $\sim$0.01, $\sim$0.1, and $\sim$0.3 atoms g$^{-1}$ yr$^{-1}$, their respective values under $\sim$40 m of ice. The hypothesis predicts [10]Be, [26]Al, and [14]C concentrations in the range 100-200, 900-1500, and 1300-2600 atoms g$^{-1}$, respectively (Fig. 7), and is therefore clearly falsified by the observations. Therefore, the ice sheet was thinner during the Holocene than it is now.

## 5  Ice thickness change inference from forward model for cosmogenic-nuclide concentrations

To quantify Holocene ice thickness change at the core sites, we used a constrained random search algorithm, coupled to a forward model for cosmogenic-nuclide production, to identify ice thickness change histories that fit the observations.

The input to the forward model is a history of ice surface elevation change at a sample site, expressed as a series of $(t, H)$ pairs, where $t$ is the time in years before present and $H$ is the ice surface elevation in meters above sea level.
Surface elevation $H(t)$ is converted to an ice and/or firn thickness $Z(t) = H(t) - H_0$, where $H_0$ is the current elevation of the sample. This procedure implicitly disregards glacioisostatic elevation change and assumes that the sample elevation is constant. It would be possible to estimate glacioisostatic elevation change by reference to model simulations or to relative sea level data from Pine Island Bay (Braddock et al., 2022) indicating approximately 20 m of relative sea-level fall during the late Holocene. However, elevation change of this magnitude would imply no more than a 2.5% variation in
cosmogenic-nuclide production rates, which is not significant.





The mass thickness of ice and firn $z(t)$ (g cm$^{-2}$), which is necessary to compute nuclide production rates when a sample is ice-covered, is obtained from the relationship:

$$z(t) = Z(t)\rho_{ice} - D\left[1 - exp\frac{Z(t)}{L}\right] \tag{1}$$

where $Z(t)$ has units of cm, $\rho_{ice}$ is the density of glacier ice (0.917 g cm$^{-3}$), and $D$ (680 g cm$^{-2}$) and L (1130 cm) are

5 constants obtained from fitting this equation to ice and firn densities measured in the field during drilling (Figure 8).

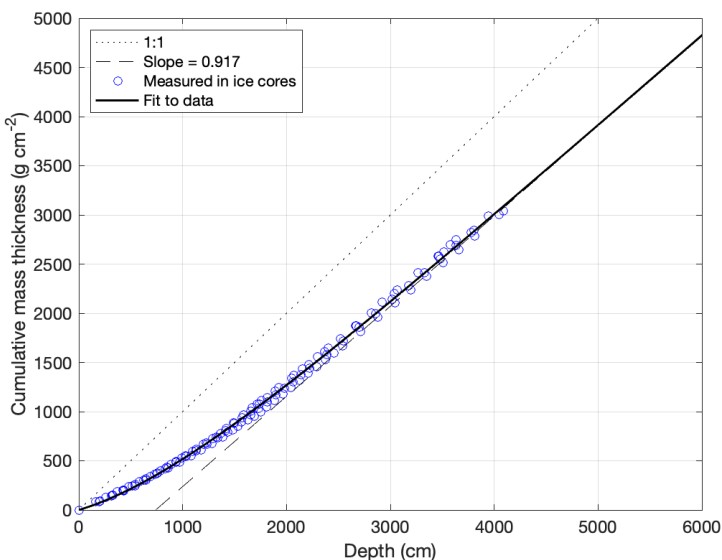

**Figure 8.** Relationship between linear thickness and mass thickness for ice and firn at Kay Peak core sites. Blue circles are observed cumulative mass thickness derived from downcore measurements of firn and ice density at all core sites. The solid black line is Equation 1 with fitted values of D and L.

Cosmogenic nuclide production rates $P(H, z)$ (atoms g$^{-1}$ yr$^{-1}$) are calculated from surface elevation $H(t)$ and mass thickness $z(t)$ as the sum of near-surface production by high-energy neutron spallation and production by deeply penetrating muons. Spallogenic production is exponential with an attenuation length of 140 g cm$^{-2}$ (Balco, 2017). Surface production rates due to spallation use the scaling method of Stone (2000) and the Antarctic atmosphere approximation of

10 Radok et al. (1996), as implemented by Balco et al. (2008). Spallogenic $^{10}$Be and $^{26}$Al production is calibrated using the 'primary' calibration data set of Borchers et al. (2016). Spallogenic $^{14}$C production is calibrated from measurements at Tulane of the $^{14}$C concentration in the "CRONUS-A" intercomparison standard (Jull et al., 2015), which was collected from a site in the Antarctic Dry Valleys, and the assumption of production-decay equilibrium at that sample site (Balco et al., 2019; Goehring et al., 2019; Nichols et al., 2019). Production rates due to muons use "Model 1A" of Balco (2017) with



cross-sections for $^{10}$Be, $^{26}$Al, and $^{14}$C production as calibrated in that reference. Given the production rate $P(H(t), z(t))$, the nuclide concentration at the present time is the integral:

$$N = \int_{0}^{t_{max}} P(H(t), z(t) + z_s)e^{-\lambda t}dt \qquad (2)$$

where $z_s$ is the mass depth of a core sample below the bedrock surface (g cm$^{-2}$) and $\lambda$ is the decay constant (yr)

for the nuclide in question. Samples are assumed to have a zero nuclide concentration at the beginning of the elevation change history ($t = t_{max}$). For computational efficiency, $P(H, z)$ is precalculated on a 2-d mesh in $H$ and $z$, and used as a lookup table during integration. Interpolation in $z$ is log-linear to improve accuracy. We used the default numerical integration routine provided by MATLAB numerical computation software.

The random search algorithm generates piecewise-linear age-elevation histories $H(t)$ that are constrained by ge-

ological and geochronological observations, applies the forward model for cosmogenic-nuclide production to compute predicted nuclide concentrations in the core samples corresponding to the age-elevation history, and then compares predicted to observed concentrations to identify $H(t)$ that fit the observations. The randomly generated age-elevation histories have the following properties. First, the ice surface is at 350 m elevation between 35 ka and the early Holocene (Johnson et al., 2020). 350 m is chosen because it is slightly above the elevation of the highest glacial erratics with

post-LGM exposure ages found on Kay Peak Ridge (Johnson et al., 2020). The LGM surface may have been higher, but the exact value chosen is not significant in this calcualtion because cosmogenic-nuclide production rates under > 100 m of ice are insignificant in our calculation. Second, the beginning of deglaciation is drawn from a uniform random distribution between 9-6 ka. These limits are selected to be consistent with regional exposure-age data (Johnson et al., 2020; Adams et al., 2022). Third, the elevation change history after initial deglaciation consists of three piecewise-linear

segments between the beginning of deglaciation and the present time. These are generated by selecting two times from a uniform random distribution within this interval, and two surface elevations from a uniform random distribution between the elevation of the top of sediment-rich ice at a core site (0.7-1.2 m above the bedrock surface) and 100 m elevation (20-25 m above bedrock at the core sites). The 100 m upper limit was chosen after initial experiments to exclude regions of the parameter space that would be certain to yield poor fits to the data, thus improving the efficiency of the random

search algorithm. Constraining the minimum thickness to be the thickness of sediment-rich ice observed at present is based on the assumption that the dirty basal ice is the result of subglacial erosion at the LGM (Stone et al., 2003; Balco et al., 2013), and if the ice had been thinner following the LGM, this unit would no longer be present. Fourth, the ice at the core site reaches its present (2019-20) elevation at $t = 0$. Omitting post-1966 thinning to simplify this constraint does not have a significant effect on the results.





We quantify the fit of predicted to observed $^{10}$Be and $^{14}$C concentrations in cores using an error-weighted sum-of-squares statistic M:

$$M = \frac{1}{n} \sum_{i=1}^{n} \left[ \frac{(N_{m,i} - N_{p,i})}{\sigma N_{m,i}} \right]^2 \tag{3}$$

where $N_{m,i}$ is the measured nuclide concentration (atoms g$^{-1}$) in analysis $i$, $N_{p,i}$ is the nuclide concentration pre-
dicted by the forward model for the sample on which analysis $i$ was made, and $\sigma N_{m,i}$ is the uncertainty in the measured
nuclide concentration. This formulation implies a normal uncertainty distribution for $N_{m,i}$. As this is not the case for the
$^{14}$C measurements (Supplementary Methods 2), we approximate the uncertainty distribution for $^{14}$C measurements as
an asymmetrical normal distribution with different values of $\sigma N_{m,i}$ above and below. Because $^{26}$Al measurements were
only made in one core (19-KP-H1), we use only $^{10}$Be and $^{14}$C data in computing M, thus standardizing the calculation
for all cores. However, as discussed below, $^{26}$Al data are exactly consistent with the $^{10}$Be measurements, so including
or not including them in calculating M for core 19-KP-H1 does not affect the results. If we assume that all measurement
uncertainties are correctly characterized, the model correctly represents all operative processes, and predicted and ob-
served values differ only due to measurement uncertainties, then the expected value of $M$ is near 1. However, these
assumptions are not completely correct. For example, we have disregarded uncertainty in production rates due to muon
interactions, which may be 10% or greater (this was omitted because production rate uncertainties are correlated be-
tween core samples, which complicates interpretation of a misfit statistic). Thus, we view minimum values of $M$ near 2
generated by the random search algorithm as acceptable fits. All results of the random search algorithm are shown in
Supplementary Methods 3.

Ice thickness change histories generated by the random search algorithm that fit the bedrock core data uniformly
include a period of several thousand years during which the ice was several meters thick at the core sites (Figs. 9, 7;
Supplementary Methods 3). Ice thickness during this lowstand is in the range 2-5 m, 3-6 m, and 4-7 m at 19-KP-H1, 19-
KP-H4, and 19-KP-H5 respectively, implying 30-35 m of thinning relative to present. Although we only assessed the model
fit against data from the bedrock cores and not against the exposure-age data from the exposed ridge, scenarios that
fit the subsurface data are consistent with the older bound of the surface data, which is consistent with the hypothesis
that scatter in these data is the result of locally derived snow- and icefields that persisted after ice surface lowering.
Scenarios with shorter lowstands, or thicker ice during the lowstands, predict systematically lower cosmogenic-nuclide
concentrations than observed (Fig. 7).

Ice thickness change histories that fit the measured nuclide concentrations display a tradeoff between lowstand duration
and ice thickness during the lowstand. A longer lowstand duration predicts higher nuclide concentrations, and thicker ice
during the lowstand predicts lower nuclide concentrations. A longer-duration lowstand with thicker ice therefore yields a
similar fit to data as a shorter-duration lowstand with thinner ice during the lowstand. Thus, it is not possible to choose
a single best-fitting thickness change history or to accurately date the beginning and end of the lowstand. For example,
a 4000-year period with 3-m-thick ice fits the data for core H1 as well as a 7000-year period with 5-m-thick ice (Fig. 9.).





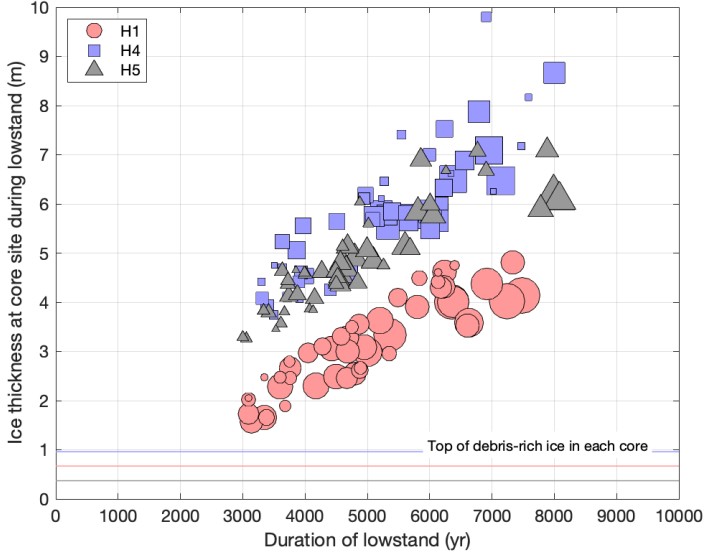

**Figure 9.** Tradeoff between lowstand duration and ice thickness during lowstand in fitting measured $^{10}$Be and $^{14}$C concentrations. This shows the 50 (out of 2000) best-fitting iterations of the random search algorithm applied to each core, where the ice thickness history in each iteration is simplified by representation as the duration of a late Holocene ice thickness minimum and the mean ice thickness during that period. Relatively long lowstands with relatively thick ice fit the observations similarly as relatively short lowstands with relatively thin ice. The size of the symbols reflects goodness-of-fit: scenarios represented by larger symbols have lower values of $M$. For best readability, symbol size is scaled to the respective range of $M$ in each core. This highlights that scenarios with longer lowstand durations are slightly, but not strongly, favored. This figure also shows that ice thickness histories that fit data from each core are systematically offset: no matter the duration of the lowstand, ice covering H1 is required to be ∼2 m thinner during the lowstand than ice covering H4 and H5.

However, all cores require a lowstand period no less than 3000 years long during which ice was no thicker than several meters (Figs. 7, 9).

Finally, the fact that the same late Holocene lowstand scenarios are inferred from fitting $^{10}$Be, $^{14}$C, and $^{26}$Al measurements separately (Fig. 10) precludes the possibility that any significant fraction of the measured $^{10}$Be or $^{26}$Al inventory was inherited from pre-Holocene exposure, perhaps in an interglacial period prior to the LGM. If this were the case, corresponding inherited $^{14}$C would have been removed by radioactive decay, and a longer period of exposure would be required to fit the $^{10}$Be and $^{26}$Al data than the $^{14}$C data. Subglacial erosion during the last glacial cycle, as also suggested by the presence of debris-rich basal ice and IRSL data precluding sunlight exposure in the last 200-300 ka, likely removed rock surfaces exposed during prior interglaciations.





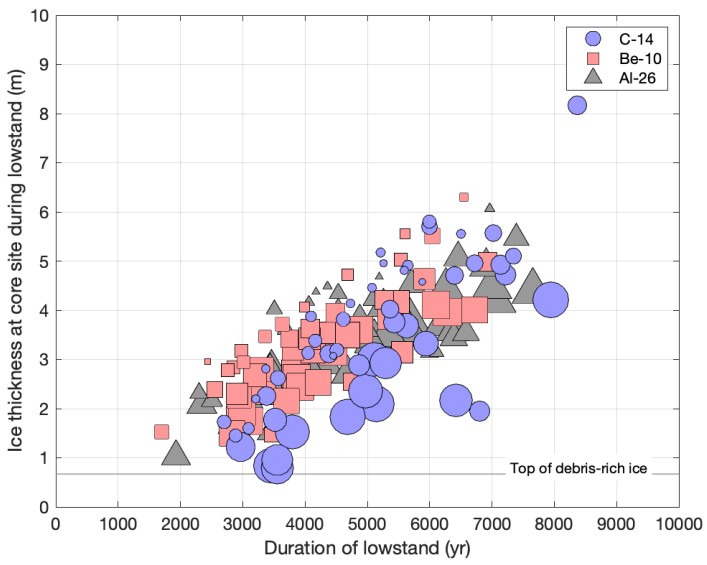

**Figure 10.** Elevation change histories that fit $^{10}$Be, $^{26}$Al, and $^{14}$C data sets, considered separately, for core 19-KP-H1. As in Fig. 9, each elevation change history is represented by a lowstand duration and a mean ice thickness during the lowstand. The best-fitting 50 out of 1000 results are shown for each nuclide, and, again following Fig. 9, symbol size is proportional to goodness-of-fit. Fitting to $^{10}$Be and $^{26}$Al measurements separately yields identical results. Although a wider range of lowstand thicknesses are permitted by the $^{14}$C data because they have larger measurement uncertainties and therefore more scatter, concentrations of all three nuclides require a lowstand of similar magnitude and duration.

To summarize, although IRSL data show that bedrock at the core sites was not ice-free in the Holocene, the cosmogenic-nuclide data show that ice at the core sites was 30-35 m thinner than present for at least 3000 years, and possibly as long as 8000 years, during the middle to late Holocene. The coincidence of IRSL data that preclude direct exposure of rock surfaces to sunlight and cosmogenic-nuclide data that require significant ice thinning is consistent with the present 5 geometry of ice and rock on the exposed portion of Kay Peak Ridge. Locating the ridge axis using single-channel ice-penetrating radar, as we did, is not expected to be precise at meter scale because of limits on instrument resolution and the likelihood of off-axis reflections from irregular bedrock topography. The currently exposed portion of the ridge is mantled by icefields on both sides that extend to within meters of the ridge axis and thicken with distance from the axis (Fig. 2). In addition, lowstand ice thicknesses inferred from the random search algorithm vary between cores over a range 10 of 2-3 m (Fig. 9), but the core top elevations vary over a range of 6 m. These observations suggest that meters-thick ice present during the lowstand does not reflect a flat ice surface uniformly 35 m below the present surface, but instead reflects a condition like what we observe now, in which fringing icefields of varying thickness covered drill sites that are




slightly off the ridge axis. This in turn indicates that the estimate of 35 m of thinning is a minimum. Areas farther from the ridge axis likely thinned more.

## 6 Summary and conclusions

Our subglacial bedrock exposure-dating results are direct, unambiguous evidence for ice thinning and subsequent thick-
ening of at least 35 m during the past several thousand years. By analogy with present conditions, thinning of this magnitude may have been associated with grounding line retreat of tens of kilometers upstream of present locations. This evidence, the association of recent ice surface lowering at Kay Peak ridge with recent retreat of the Pope Glacier grounding line, and independent evidence for falling relative sea level in the nearby Pine Island Bay region during the past 4000 years (Braddock et al., 2022), together indicate that a grounding line retreat-advance cycle driven by glacioisostatic
rebound–RSL feedback took place during the Holocene. In general terms, this result provides supporting evidence for the hypothesis that glacioisostatic rebound can provide an important stabilizing feedback on grounding line retreat. How-ever, boundary conditions during a late Holocene retreat-readvance cycle were not identical to present conditions. Thus, whether our results, or other evidence for Holocene retreat-readvance and/or RSL fall elsewhere in Antarctica (Bradley et al., 2015; Kingslake et al., 2018; Venturelli et al., 2020; King et al., 2022), can be used to understand the importance
of glacioisostatic rebound feedback in stabilizing present grounding line retreat in the Amundsen Sea region depends on (i) the difference between isostatic responses to large-scale LGM-to-Holocene thinning that took place over thousands of years and thinning of lesser magnitude observed in recent decades; (ii) the relative importance to grounding line position of relative sea level change, ice shelf buttressing, and sub-ice-shelf melting driven by oceanographic conditions, which cannot be deduced from any data presented here; and (iii) past and present bed topography in the region of grounding
lines. In addition, the past thinning-thickening cycle that we have detected spanned a duration of at least 3000 years, which is geologically rapid but slow by comparison to the timescale of projected sea-level rise impacts of present ice sheet thinning.

*Code and data availability.* All data described in this paper that have not already been published elsewhere are included in the sup-
plementary data. MATLAB code used to carry out all calculations in this paper is included in the supplementary data for purposes of paper review only, and is *not at this time licensed for any other use, reproduction, or distribution*. In the event of paper acceptance and publication, the code will be posted on a publicly available repository under an open-source license.



*Author contributions.* Author contributions following the CRediT Authorship Guidelines:

| Author: | JA | GB | NB | SB | SC | BG | BH | JJ | KN | DR | RV | JW | KW |
|---|---|---|---|---|---|---|---|---|---|---|---|---|---|
| Conceptualization | | X | | | | X | | X | | X | | X | |
| Methodology | | X | X | | | X | | X | X | X | X | X | |
| Software | | X | X | | | | | | X | | X | | |
| Validation | | X | X | | | X | | X | X | X | X | | X |
| Analysis | | X | X | | | X | | | X | X | X | | X |
| Investigation | X | X | X | | X | X | | X | X | X | X | | X |
| Resources | | X | | | X | X | | X | | X | | | X |
| Data curation | | X | X | | X | X | | | X | X | X | | X |
| Original draft | | X | X | | | | | | X | | X | | |
| Review/editing | X | X | X | X | X | X | X | X | X | X | X | X | X |
| Visualization | | X | X | | X | | | | | | | | |
| Supervision | | X | | | | X | | X | | X | | | |
| Administration | | X | | | | X | | X | X | X | X | X | |
| Funding acquisition | | X | | | X | X | X | X | | X | | X | X |

*Competing interests.* The authors declare no competing interests.

*Acknowledgements.* This work would not have been possible without the efforts of: many members of the U.S. Antarctic Program and the British Antarctic Survey, in particular Chris Simmons, Leslie Blank, and Nick Gillett; pilots and crew of Kenn Borek Air, Ltd.

5 and the 109th Airlift Wing of the New York Air National Guard; the U.S. Ice Drilling Program, in particular Grant Boeckmann and Eliot Moravec; the Polar Geospatial Center of the University of Minnesota; and the National Ocean Sciences AMS facility, in particular Mark Roberts. This work is part of the Geological History Constraints project, which is a component of the International Thwaites Glacier Collaboration (ITGC), and was supported by the US National Science Foundation (grants OPP-1738989 and EAR-1806629), the UK Natural Environment Research Council (Grants NE/S006710/1, NE/S00663X/1 and NE/S006753/1), and the Ann and Gordon Getty

10 Foundation. The U.S. Ice Drilling Program is supported by NSF Cooperative Agreement 1836328 and the Polar Geospatial Center by NSF grant OPP-2029685. This is ITGC Contribution no. 66.





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
