# Peer review of "Reversible ice sheet thinning in the Amundsen Sea Embayment during the Late Holocene"

_The Cryosphere, 2022_

## Author Comment (AC1)

**Response to review 1 (Jason Briner) of 'Reversible ice sheet thinning in the Amundsen Sea embayment during the late Holocene', by Balco, Brown, Nichols, Venturelli et al.**
* * *
Nov 21, 2022

First, we would like to thank Jason Briner for a supportive and helpful review. We will respond to all the review comments (reproduced in *italic*) in order, and then at the end we will provide a summary list of all the proposed revisions. In general, this review was quite supportive of the paper, and many of the review comments did not represent issues requiring correction, but instead contained questions or comments suggested by aspects of the paper. Thus, we have thoroughly discussed all aspects of the review in the responses, but several of the review points did not lead to any proposed revisions.

1. *The glacier thickness history most compatible with the data, according to the interpretations laid out in the paper, is one that I would characterize as actually rather stable. Following a period of rapid ice thinning between ~6000 and 8000 years ago of >200 m in <1000 years is a subsequent period 1000s of years in duration with a thinning of only "30-35 m relative to present." One could interpret this amount of glacier thickness change across many millennia as minor, and maybe not a reason to question the irreversible retreat concept, as the title does. There is mention in the conclusion paragraph that 30-35 m of thinning "may have been associated with grounding line retreat of tens of kilometers upstream of present locations." With the emphasis of the paper being on the reversible nature of marine glaciers in West Antarctica, I might suggest the addition of evidence/discussion backing up the inference of 10s of km of grounding line retreat associated with 35 m of thinning.*

Our response to this has two important elements. First, whether something is "stable" or not is relative and depends on not only the frame of reference, but also in many cases the rhetorical point that a paper is trying to make. For this reason, we have avoided the use of "stable" and instead characterize amounts of thinning or retreat with numerical values. Rereading the paper reveals that we did not completely succeed in this goal, having failed to remove one instance of the word 'stable' on page 6 (see additional discussion of this below). We used "unstable" once in a mathematical/physical context ("stable" and "unstable" have clearly defined meanings in mathematics) to introduce the marine ice sheet instability feedback, not to describe ice sheet behaviour. Thus, our aim in this paper is explicitly not to characterize ice sheet behaviour as "stable" or "unstable," but instead to place quantitative constraints on past ice thickness. At no time have we stated whether our reconstruction of Holocene ice sheet thinning and thickening is "stable" or "unstable."

Second, although it is true that 35 m of thinning is small compared to hundreds of meters of LGM-to-present thinning, the point of our argument is that it was followed by thickening, not by more thinning. A central reason for concern about the irreversibility of present thinning is that, due to reversely-sloping subglacial topography and the marine ice sheet instability, grounding line positions inboard of present may not be stable (in the mathematical sense), so grounding line retreat inboard of present may result in irreversible retreat into the center of the ice sheet. The importance of our results is that if past thinning below present was also associated with past grounding line retreat inboard of present, as we infer from coeval thinning and retreat in recent decades, then grounding line positions inboard of present did not lead to irreversible retreat.

Although we discussed these points in Section 1 (p. 2, lines 5-10, p. 5, lines 1-10) and then again very briefly in Section 6 (p. 20, lines 5-10) we did not lay out this entire chain of reasoning in one place. To some extent, we did not highlight this reasoning because the aim of this paper is to present observational data, and not to speculate about the importance of the observations. However, we can clarify this reasoning by adding discussion to these sections of the paper to more clearly lay out the reasoning that

(i) grounding line positions inboard of present are potentially of concern, and (ii) the evidence for mid-Holocene thinning indicates that the mid-Holocene grounding line position was likely inboard of present (**Revision 1**).

- *I think that an underlying requirement for applying luminescence exposure dating in this situation is that a sub-glacial site of interest experiences(ed) non-erosive glaciation. If the base of the glacier is erosive, and the drill sites had been exposed in the middle Holocene, the glacier could have eroded the upper few of cm of the bed during late Holocene overriding. In this case, wouldn't the luminescence results would be the same as if the drill sites were not exposed at all during the Holocene (the current interpretation)?*

This review comment, as well as the subsequent ones, highlight an omission in the paper, for which we apologize. In fact, ice at the drill site could not have been above freezing during a late Holocene thickening event with a magnitude of tens of meters, so erosion could not have occurred. Mean annual temperatures at the drill site are well below freezing throughout the year, and hundreds of meters of thickening would be required to bring the ice-rock interface above freezing.

There is evidence from similar sites around Antarctica that hundreds of meters of LGM thickening did result in ice above the freezing point at low elevations that are near the present ice margin. Such observations are described in, among others, Stone et al. (2003, Science) and Balco et al. (2014, QSR). At these sites, geomorphic evidence of erosion by wet-based ice (polish, striations) is observed only at the lowest currently exposed sites that are hundreds of meters below the LGM ice surface. Cosmogenic-nuclide data show concordant bedrock-erratic ages only at these lowest sites. At all higher elevations, cosmogenic-nuclide concentrations are higher in bedrock, and erratics with ages predating Holocene deglaciation are present, indicating cosmogenic-nuclide inheritance and therefore frozen-based conditions at the LGM. It is likely that the ice margin at Kay Peak Ridge is near a similar transition: Be-10 data on boulders and bedrock above the present ice margin display inheritance and therefore a lack of LGM erosion, whereas much lower Be-10 concentrations attributable to only Holocene exposure in core tops indicate likely LGM erosion at the core sites. Regardless, Be-10 inheritance near the present ice margin indicates that erosion was not widespread here even at LGM and early Holocene conditions with hundreds of meters thicker ice. Thus, tens of meters of late Holocene ice cover could not result in erosion at the core sites. Because erosion and sediment transport at the bed could have occurred during LGM conditions, we attribute the debris-rich basal ice to the LGM.

The point here is that these observations are, in fact, critical to understanding why the core tops could not have been ice-free during the Holocene, but we omitted to discuss them. Presumably, this is because we mostly work in Antarctica and it seemed obvious to us. However, we can clarify this in a revised manuscript by adding material at the end of section 2. After we have clarified that the ice-rock interface is, in fact, frozen, and described the debris-rich ice in contact with the bedrock, we can discuss the constraints on basal temperature and our reasoning as to why the debris-rich ice can only have formed during LGM conditions. (**Revision 2**)

- *The authors might therefore mention that the drill sites are cold-bedded (I didn't find anywhere in the paper that mentions this, but I might have missed it). I believe Winke operations require a cold bed at the time of drilling? (not sure about the packer set-up, if that could still obtain bedrock cores with minor drill fluid loss at a warm ice-bed interface) Knowing the basal temperature would be best (but maybe not possible). If the bed is only slightly below pressure melting, then it may be difficult to know if it was always cold-bedded during the Holocene. It might come across as a surprise to some that this is a site of cold-bedded glaciation, given the marine-based, fast-flowing ice-steam nature of these glacial systems; literature suggests*

*velocities on order 1 km/yr and beds nearby that are not frozen, although there must be sharp boundaries in a landscape like this.*

See above. In addition, although certainly fast-flowing ice is present in major glaciers near the site, fast-flowing regions of these glaciers are hundreds to > 1000 m thick. The core site is not within a region of fast flow at present (although it may have been closer during LGM conditions).

Also, yes, the drill system requires a frozen bed. We will clarify this (**Revision 2**)

- *The authors could attempt to rule out another possibility for the OSL signal to be saturated at the surface during an ice-free period, which would be that the two drill core sites with the OSL data were covered by debris during a Holocene ice-free period. Such debris could shield underlying bedrock from being bleached. If this is unlikely, say why. Perhaps more information from the ice-free portion of the bedrock ridge would be useful – does it lacks debris, or is debris sparse? Similarly, back to the glacial erosion part, maybe some descriptions from this ridge would be useful here as well. Does it lack glacial molding, other glacial-erosional landforms and striations (particularly below the 1966 limit)?*

Again, we address this in the discussion above. If, as we conclude, the debris-rich basal ice can only have formed under LGM conditions, then the core tops could not have been ice-free during the Holocene.

- *Given the two reasons above that could lead to a saturated OSL signal despite an ice-free middle Holocene, it could be worth mentioning whether the CRN results alone are or are not compatible with an ice-free period during the Holocene. If not, then great, and the interpretation in the present manuscript is further bolstered.*

The Be-10 and C-14 concentrations, no matter what they were and even if they were indistinguishable from blank, could always be compatible with a very short ice-free period. For example, ice-free conditions for one year would produce about 15 atoms/g of C-14 and 5 atoms/g of Be-10 at the rock surface, which would be undetectable. Even though the random search inversion rejects thinning histories that have less than a couple of meters of ice for a couple of thousand years, it would always be impossible to disprove the hypothesis that there was a short ice-free period that was too short to produce a measurable amount of Be-10 or C-14. On the other hand, an ice-free period as short as hours or days would significantly perturb the OSL signal. This is why the combination of cosmogenic-nuclide and OSL data, which have contrasting (integral vs. instantaneous) responses to exposure, is so useful in this application.

- *The authors pioneered the 14C/10Be chronometer for Holocene ice burial. I might have missed it, can there be mention of this ratio in the bedrock cores and whether it or supports or excludes a scenario of ice-free conditions followed by ice burial? Are there any 10Be measurements from the ridge between today's ice height and the 1966 limit? Having a C/Be ratio in this portion could be informative in constraining the pre-1966 duration of ice at this location.*

We intentionally did not mention the 14/10 ratio in the core samples. There are two reasons for this. First, the model is fit to the direct measurements (the concentrations), not to a derivative of the measurements (a ratio). Because the ratio is just computed from the concentrations, the ratio cannot support or exclude anything that the concentrations by themselves do not support or exclude. Second, in most applications of surface exposure dating, the purpose of calculating a ratio of two nuclides is to plot data on a commonly-used exposure-burial diagram (the so-called 'banana diagram'). The purpose of this diagram is that if one assumes that a sample has experienced a single period of surface exposure followed by a single period of burial, the diagram allows graphical estimation of the duration of these two events. However, an important element in constructing this diagram is that measured concentrations in a sample must be normalized to a surface production rate so that they can be compared to idealized isochrons of

exposure and burial duration, which are constructed by assuming a nuclide production ratio at the surface. This is an effective way to represent data that were collected from rock surfaces. However, it is not effective for representing data collected below the rock surface, because nuclide production ratios vary with depth. Each data point in a depth profile must not only be plotted as a separate point on a two-nuclide diagram, but must also be compared with a separate, distinct set of exposure and burial isolines. This leads to a nearly uninterpretable diagram. In addition, in this work, because we expect that the thickness of ice covering a core site is changing through time, we do not even know in advance the depth at which production in our samples took place, so we cannot even construct the burial-exposure isochrons. A burial-exposure diagram without any burial-exposure isochrons would be meaningless. Another way of saying this is that if we don't know in advance what the exposure depth is, we don't know what the production ratio is, and if we don't know what the production ratio is, then we can't interpret the measured ratio to mean anything by itself. For these reasons, discussion of the 14/10 ratio in the subsurface samples would most likely be misleading rather than illuminating, and we have intentionally avoided discussion of the ratio.

The paper does include paired 14/10 data for two bedrock samples above the modern ice surface. What the review is proposing is that these data could be used, as they have been in some situations from alpine glacier forelands in temperate regions, to infer durations of Holocene exposure and burial. However, in our case the apparent Be-10 ages are pre-Holocene, indicating that Be-10 inheritance is present, so the 14/10 ratios yield no information about Holocene events. In the alpine glacier examples, all inheritance was removed by subglacial erosion, so two measurements (Be-10 and C-14 concentrations) can be used to solve for two unknowns (length of Holocene exposure, length of Holocene burial). In our case, there is an additional unknown (the amount of Be-10 inheritance), but still only two measurements, so the solution is undetermined.

- *The exposed bedrock below the 1966 limit is a playground and offers much. It is unknown when this portion of the ridge was re-occupied by ice, but I believe it is thought that the episode of thinning earlier in the Holocene led to the exposure of the portion of the ridge between the 1966 level and the present ice surface – and the 14C numbers reflect this. Does the recent ice cover (unknown period preceding 1966) need to be considered when interpreting the 14C ages of bedrock in this zone? I guess the 14C ages here are apparent ages? Also, is there any evidence of glacial abrasion in this zone? If so, I suppose some or all of it could date to the LGM. But if all of it doesn't date to the LGM, could the 14C age scatter (in the paper explained by 'locally derived snow and ice') be explained by minor amounts of uneven late Holocene glacier erosion of the bedrock surface?*

The data from this part of the ridge are not as informative in this regard as this comment suggests. Because Be-10 concentrations indicate pre-LGM inheritance (see above), paired 14/10 data on bedrock provide no information about Holocene events. In addition, there are no properties of the data set from this part of the ridge that require late Holocene cover. For example, in a similar situation, Balco et al. (2012, QSR) observed both mid-Holocene and extremely recent (~100 yr) exposure ages in an elevation range known to have been covered by ice in recent decades. In that case, late Holocene thickening could be inferred from the exposure-age data; the two co-occurring populations are not easily explained without a multistage exposure history. At Kay Peak, however, there are no properties of the exposure-age data that could not be explained by continuous exposure since the mid-Holocene: the age distribution and scatter are similar from the ice margin to 250 m above, without evident discontinuities. In general terms this would tend to indicate that the pre-1966 period of ice cover of this site was short relative to the C-14 half-life (hundreds of years rather than thousands), but can't be interpreted more quantitatively without additional assumptions.

And, again, bedrock erosion under tens of meters thicker Holocene ice is not glaciologically feasible, as discussed above.

*Finally, are the CRN depth profile shapes themselves of any use in determining where in depth space the drill cores are – ie, has there been any surface truncation? Some of the profile data (H4 and H5) look remarkably flat, that is, fairly unchanging with depth (in Fig 7A-H5, the statistical profiles have a rather uncomfortable fit with the data). Perhaps this is a product of residing below meters of ice and a few cm of surface erosion would not be possible to detect.*

The answer to this is yes, the shape of the depth profiles is critical in fitting the model. Basically, the model fitting is seeking to do two things: match the concentrations and the shape of the profile. This explains why the best-fitting model results display a tradeoff between lowstand duration and ice thickness. Increasing the ice thickness during an ice-free period increases the effective attenuation length, but decreases the production rate, so thicker ice requires a longer lowstand. If there was no depth profile and only a surface sample existed, then any length of lowstand could be fit by adjusting the ice thickness, and any ice thickness could be fit by adjusting the length of the lowstand. It would not be possible to constrain either parameter. We explain this relationship on pp. 17-18 in the original MS, and it has also been discussed at length in other references, for example Schaefer et al. (2016, Nature).

3. *As I wrote in my intro remarks, this manuscript does a remarkable job explaining a complicated dataset simply. The figures are outstanding. Yet, they are mostly highly technical. This is, of course a technical paper in a technical journal. Nevertheless, I wonder about a summary figure that shows the glacier history in a more cartoon fashion. A multi-panel cross section or something that shows the ice thickness history in several time slices through the Holocene. Something for the broader readership. Additionally, there seems to be some important interpretations that are rather nuanced and difficult to visualize without such a figure, such as the potential that the drill sites are slightly off ridge and could be covered by a ramp of glacier ice. This is rather important, as this point is used to suggest that the calculated ice thinning may be a minimum. Finally, some field pictures of sample sites could be really helpful, and what about any photographs of the cores themselves? For those of us who like to teach about very cool studies like this, these visualizations go a long way.*

Although we appreciate this point, a cartoon or visualization figure that aimed to show the ice sheet configuration at various times in the Holocene is problematic in the context of this paper, because drafting such a figure would by necessity involve extrapolating the reconstructed ice thickness history at the core sites to other locations. What we have done in this paper is reconstruct the ice thickness history at one site and one site only. This is extremely important because the significance of our reconstruction depends on how one extends this observation to other locations. Specifically, does mid-Holocene thinning at this site imply regional grounding line retreat, as the observations from recent decades imply? Or is thinning at this site purely a local effect that is unrelated to the grounding line position? Because this question is so important, we have intentionally been extremely clear in this paper that we have reconstructed ice thickness changes at a single site only. This reconstruction is a major accomplishment, because it has never been done before in a targeted bedrock recovery drilling project. However, the larger significance of this result depends on how the observations from one site are related to conditions elsewhere. The only evidence we have that addresses this question is the association between thinning at the site and grounding line retreat observed in recent decades. This evidence does imply that mid-Holocene thinning was also associated with grounding line retreat, so we make this point and then stop. The next step in evaluating the implications of our results for the regional ice sheet configuration would be a high-resolution modeling study, which is not part of this work. For this reason, we have specifically

presented the results of this work as an ice thickness change history at one site (as shown in Figs. 7, 9, and 10), and not as a reconstruction of the ice sheet configuration in nearby areas. Thus, we conclude that a cartoon or visualization figure would be potentially misleading.

Finally, with regard to photos of the cores, it is true that these cores are notable because there are only a few cores that have ever been collected from bedrock beneath ice sheets, but from the geologic perspective they are not very exciting – they are completely normal-looking cores of unremarkable biotite gneiss that look exactly like the outcrop on Kay Peak Ridge a few hundred meters away. There's nothing about the appearance of the cores that is significant to the paper in any way, so there does not appear to be any value in showing photos of the cores in the main text. As noted, the supplementary data contains links to the core photos, but we could certainly include the photos themselves in the supplement.

*- page 2 line 9, could be useful not only to mention that people have modeled irreversible glacier retreat systems, but what they have found.*

We can clarify this section (**Revision 3**)

*- page 2 line 20, here the authors are describing how OSL-ED and CRN systems work in the remote case of long-lived thin ice cover. Why not start by describing the more obvious case of an ice-free then ice-covered system. The text at present certainly foreshadows your interpretations, but sort of skips the basics first.*

Well, we were trying to make the point that OSL can't detect a period of thin ice cover, but cosmogenic-nuclide data can. Clearly this needs some revision. We will clarify this section. (**Revision 4**)

*- figure 1, add legend of the ice velocity. Also, sometimes authors use hotter colors for high velocity (reds) instead of "colder" colors like blue.*

The ice velocity is intentionally shown in receding, "cool," colors because the purpose of the figure is to show grounding line positions, and the ice velocity is only there to highlight where the glaciers are. The magnitude of the ice velocity has no relevance to the paper subject matter. We will make clear in the caption that the ice velocity is presented in a nonquantitative way. (**Revision 5**)

*- figure 2, I struggled a little bit here. The pink dots shown on the graph are fewer in number than the pink dots on the image - I'm guessing that the plot doesn't encompass the entire transect shown in the image. But I had to stare and dig deep to conclude that. Also, there is mention of a sample at 171 m asl, but this plot doesn't seem to have a dot at that exact elevation.*

As foreshadowed in the first part of this review comment, the 171 m sample is actually off scale to the left on the inset axes, so not visible. We will make clear in the caption that the inset axes only include the lowermost portion of the ridge (**Revision 6**). Note that all the elevations of erratic and bedrock samples above the ice surface are correctly represented in Figs. 4 and 7, so it is not necessary for the reader to rely on Fig. 2 for this information.

*- page 6 line 21, here it states that stable ice for millennia is unlikely; isn't that more or less the interpretation that the manuscript goes with? The ice-thickness histories plotted in Fig 7C during the middle Holocene (the middle segments) appear flat and pretty unchanging.*

As discussed above, we made a mistake and used "stable" here. What we mean to say is that the hypothesis that the ice thickness was EXACTLY THE SAME between the mid-Holocene and the present is unlikely. We will fix this. (**Revision 7**)

*- page 8 line 35 – could this "fragmented rock" be surface debris? Back to above comment about shielding the underlying bedrock surface during an ice-free period. This debris could theoretically be ephemeral, exist during an ice-free period but then be transported off a rock core site during subsequent overriding.*

Well, we don't know what it is. As the bedrock is densely jointed, the boundary between "bedrock" and "surface debris" is a bit arbitrary. However, this clast was embedded in debris-rich ice, not in contact with the bedrock, so it could not have been resting on the surface. We will clarify that whatever it is, it was embedded in the debris-rich ice and not in contact with bedrock. (**Revision 8**)

*- figure 6, fantastic idea to re-occupy an older rock sampling divot. Could you model a profile that shows a plausible history alternative to the one this paper points to? - that is, middle Holocene exposure until something like 2 ka, and then subsequent burial to present? It could be nice to see such a scenario stand in contrast to these data that were generated, much in the way that the null hypothesis is plotted in Figure 7A/B. Also, as an aside, it sure would be cool to have a core with OSL data from the above-1966 ridge just to see if you can match the 14C exposure ages – would be a nice proof of concept.*

Figure 1 below shows this model calculation. This scenario would predict $L_N/T_N$ signals very different from what we observed, and can be easily excluded.

[Figure]

Figure 1. Observed $IR_{50}$ (blue boxes) and $pIRIR_{225}$ (red circles) luminescence depth measurements from core top sample 19-KP-H1 are compared to depth profiles from a hypothetical, simulated exposure history in which the rock surface was completely exposed during a Holocene ice thickness lowstand. In this history, the sample was exposed to sunlight for 5 kyr and then buried by ice for 2 kyr before measurement. This highlights that the measurements are incompatible with this scenario. The two panels show the same information on logarithmic and linear y-scales.

*-page 12 line 11, not sure I entirely understand this sentence. It says that the 14C "concentrations are 2-3 times typical detection limits," but then also "and are significantly lower than routinely measured." Lower on line 25, it says these samples are "near detection limits." Collectively, these statements leave me a little confused.*

We can and will clarify this. Note that we discuss the issue of the C-14 blanks at length in the response to the other review by Nat Lifton. (**Revision 9**)

*- figure 7C, is there a reason to choose the history ending with the present ice thickness at H5 in the scenario-modeling instead of a higher surface that could correspond to the "1966" thickness, which is 30 m higher? I suppose all you could do is estimate. Perhaps it is of no real consequence.*

The reason is that it would add complexity without affecting the results. Because of the need to match both the long apparent attenuation (by making the ice thicker during a lowstand) as well as the concentrations (by making the lowstand longer), the random search optimizer nearly always chooses thickness change histories in which thickening from the lowstand back to the modern elevation happens at the latest possible moment. This is evident in Figure 7: nearly all the well-fitting models shown in green come up to the modern ice surface extremely steeply at the end. The result of this is that if we force the models to come up to the 1966 elevation and then back down in 2020, the period before 1966 during which the ice is thicker than present becomes so short that it is negligible in relation to the total nuclide concentrations. Thus, adding this constraint to the model makes it more complicated but doesn't affect the results.

*- page 16 line 26, interesting interpretation of the basal debris-rich ice. It sort of seems that erosive ice during the Holocene is ruled out, therefore this debris must relate to the LGM. But I didn't find anywhere in the paper where this is stated. Ruling out more firmly a Holocene origin could help to bolster the interpretation in the paper.*

This issue is (hopefully) thoroughly discussed above.

*- page 20 line 6, here there is a lot packaged into one sentence. This amount of grounding line retreat is somewhat critical for justifying the paper's title which challenges the irreversible glacier concept. I'm not against the authors' choice to not include a lengthy "implications" portion to this paper, but they could add some supporting discussion here.*

As discussed above at more length, this is an important element of the paper. We have just made observations at one site. It is possible to make a reasonable hypothesis about how to extend these observations to a wider area by comparing them with observations in recent decades. Anything after that would be speculative without a serious modeling study. Regardless, this section clearly needs some clarification. We will improve this in a revised MS. (**Revision 1**)

*- page 20 line 9, similarly, a lot is packaged into the sentence on the RSL-rebound control on the grounding line in this system, readers might find bit more on this useful.*

We can clarify this. (**Revision 10**)

*- page 20 line 12, I generally try to avoid making stylistic/subjective comments about writing, but I'll make one here – sentence beginning with "Thus" is lengthy and I found it a little difficult to follow.*

We can clarify this. (**Revision 10**)

To summarize, proposed revisions to address these review comments are as follows:

**1**. Clarify reasoning about irreversibility/grounding line position vs. thinning in Section 1 and elsewhere
**2**. Explain clearly about cold-based ice/no erosion during late Holocene cover in Sections 1 and 2
**3**. Additional context for 'irreversible' in Section 1
**4**. Clarify cosmogenic-nuclide vs. OSL sensitivity in Section 1

**5**. Clarify that ice velocity representation is nonquantitative in Fig 1 caption

**6**. Clarify that profile axes are truncated in Fig 2 caption

**7**. Remove 'stable' on p. 6

**8**. Provide more context for loose rock recovered in fourth borehole

**9**. Clarify detection limit issue for C-14

**10**. Clarifications and editing for concluding remarks in Section 6

---

## Author Comment (AC2)

**Response to review 2 (Nat Lifton) of 'Reversible ice sheet thinning in the Amundsen Sea embayment during the late Holocene', by Balco, Brown, Nichols, Venturelli et al.**

Nov 22, 2022

To begin, we would like to thank reviewer Nat Lifton for his close attention to the technical details of the C-14 measurements. As the review recognizes, the C-14 data set reported in this paper is more extensive, and involves measurements at lower concentrations, than nearly any other published work. For this reason, we describe in this work a larger data set of measurement blanks than described elsewhere, and we thought very carefully about blank and background corrections and the associated uncertainty analysis. We very much welcome the opportunity to air these issues in the online discussion.

The bulk of Lifton's comments focus on these background corrections, specifically on three main questions: one, whether time-dependence of measurement blanks should be taken into account in blank corrections; two, how detection limits are quantified; and three, the importance of scatter in replicate measurements, and how it should be taken into account in data interpretation. Analysis and discussion of these issues was, in part, addressed in various aspects of the supplementary data provided with the paper. For example, several of the figures shown below are taken from a MATLAB notebook that is included as part of the supplementary data. However, for convenience for readers of the online discussion, in the first section of this response we will summarize and augment this analysis.

A secondary aspect of Lifton's review focused on production rate estimation and interlaboratory comparison via the CRONUS-A measurement standard. We will address this in the second section of the response. A third section of the review had some queries about the form of the measured concentration-depth profiles, which we will address in a third section of the response. Finally, the review pointed out several minor errors and omissions (e.g., 5730 vs 5700 +/- 30) that we will correct in a revised manuscript.

Overall, we will argue that (i) our analysis of the measurement background data and the blank correction scheme derived from this analysis are justified by the properties of the data set, and (ii) the issues the review brings up relating to CRONUS-A reproducibility are useful as context for the measurements, but fundamentally not relevant to the conclusions of the paper. However, we will not propose alterations to the main text of the paper based on these issues. Our reasoning here is that these technical aspects, although in many ways the most novel aspects of this work and potentially the most important for future similar applications of cosmogenic C-14, are highly technical in nature and, sadly, of interest only to a small readership. Because Cryosphere is an open review journal and this review response will therefore be publicly accessible with a DOI in future, the minority of readers that are interested in the technical discussion of this issue can find it here, without the necessity of adding extensive technical material to the main text of the paper itself that would potentially distract from the main point of the work. In addition, we will update the supplementary data to include the data and code needed to generate all the analysis and figures shown below.

**I. C-14 blank corrections.**

**I.1. Time-dependence of C-14 measurement blanks.**

Lifton proposes in his review comment that instead of using an aggregated distribution of measurement blanks over a long period (henceforth, the 'aggregate approach'), as we have done, it might be preferable

to use only blanks measured close in time to samples for blank corrections. The premise of the approach proposed by Lifton (henceforth, the 'close-in-time approach') is, basically, that the measurement background, as quantified by process blanks, varies systematically in time such that the time-dependent variation is larger than any non-time-dependent variation (of course, it is impossible to measure a non-time-dependent variation because blanks must be measured sequentially, but this is a good way to think about it).

[Figure]

*Figure 1. C-14 measurement blanks run at Tulane in 2019-2021. Error bars are 1-standard-error nominal measurement uncertainty, which primarily comes from the AMS measurement.*

*Note: the data and code to make all the figures in this response are included in MATLAB workbooks in the supplementary information:*

*C14_blank_time_dependence_202211.mlx*
*C14_uncertainty_experiments.mlx*

Figure 1 shows the sequence of measurement blanks run at Tulane over a 2-year period. This two-year measurement period is chosen because it begins at the time of the last significant change to the extraction system and procedure (a change from alumina to platinum boats for sample heating) and also encompasses all of the measurements (surface and core) included in this study. As a first point, it is immediately evident that the variability in the blank on any time scale greatly exceeds the internal measurement uncertainty in each blank inferred from the AMS and $CO_2$ measurements. Thus, the internal uncertainty estimate is not relevant to computing the true uncertainty in a blank subtraction scheme. Henceforth, we ignore the internal measurement uncertainty of the blanks and accept that the uncertainty distribution in any blank correction scheme must be derived from the distribution of a number of measurement blanks.

From visual inspection, some time-dependence appears to be present in these data, but the time-dependence appears to be manifested mainly as changes in the variability among blanks, expressed as an increased or decreased likelihood of observing high values. For example, blanks measured after July 2021 were mostly < 50,000 atoms. Between Jan 2021-July 2021, there were also many blanks in the same range, but they were interspersed with many other blanks approaching and exceeding 100,000 atoms. Thus, the variation in measures of the dispersion (e.g., the standard deviation) is more striking than that in central measures (e.g., median or mode). This is not like the situation for a typical application of a time-dependent blank correction in which different time periods display similar scatter but different central values. Remember, the condition that would justify a close-in-time approach is that the time-dependent variability is larger than the non-time-dependent variability. At first glance, it does not appear that this is the case.

It is possible to quantify the relative importance of time-dependent and non-time-dependent variability by considering the autocorrelation of the blank series. If time-dependent variability is important, the value of each blank should be highly correlated with the value of the previous (or next) blank.

[Figure]

*Figure 2. Autocorrelation in series of measurement blanks.*

Figure 2 shows the autocorrelation analysis. Although the correlation of blank *i* with blank *i-1* is significant ($p < 0.05$), it is weak (correlation coefficient 0.49). The $r^2$ of this relationship is 0.24, which indicates that only 24% of the variability in the blank is time-dependent. The majority of blank variability is therefore non-time-dependent.

If we use this observation as a guide to how to proceed with a blank correction scheme, it is evident that a strict close-in-time approach, for example using the average of two bracketing blanks for each sample or similar approaches, would lead to underestimation of the uncertainty in a blank subtraction scheme. We therefore reject this type of approach. This leaves two possible approaches. The more complex of the approaches would be to try to incorporate both time-dependent and non-time-dependent variability by a moving-window or moving-kernel type of scheme in which we sought to characterize, for example, the mean and standard deviation of measurement blanks within a certain time period around a sample measurement, and use those values for blank correction. We believe that this type of an approach is not strongly justified because it implicitly includes assumptions about the physical processes leading to variability in the blanks. For example, if measurement blank results were related to a "memory" of the sequence of non-blank samples that were run in the extraction system (which is very possible), then a moving-window or moving-kernel approach that was not aware of the samples could be very misleading. If we knew what the process leading to blank variability was, we could tailor such a scheme to avoid gross errors (for example, if we KNEW that blank variability was related to sample history we could use a backward-looking window). However, we don't know this, so we don't know whether a moving-window scheme would or would not result in serious errors.

The less complex of the two possibilities is to disregard possible time-dependence, and use an aggregate approach that uses blank data collected over a long period of time in which the extraction line configuration was not modified (as is the case for the data set shown here). The advantages of the aggregate approach are that (i) it is consistent with the observation that the majority of blank variability is non-time-dependent;  (ii) it uses the maximum amount of data so is more likely to correctly detect and quantify a long-tailed distribution (which is a suspected issue based on previously published C-14 data); and (iii) it does not require assumptions about the physical process leading to blank variability. The disadvantage is that we would be ignoring the fact that even though the time-dependent component of the

blank variability is small, it is not zero. As Lifton proposes in his review, if there is a period of time during which the blank distribution is different from the aggregate distribution, the blank correction will be incorrect. As pointed out in the review, the core sample measurements were made in late 2021, when the distribution of blanks appears to be lower and less scattered than the aggregate distribution. In this case, the risk is that the aggregate approach may lead to an overcorrection for blank and thus an underestimate of the true C-14 concentration in core samples. This, in turn, could lead to a spurious conclusion that C-14 was not present above background in core samples when in fact it was present, that is, a false nondetection. However, it could not lead to the opposite spurious conclusion, that is, a false detection. In the context of this study, a false detection is a more significant risk than a false nondetection.

Another way of expressing this is that, as stated in the review, if we used a close-in-time approach in which only blanks measured in late 2021 were used as a basis for blank-correction of the core data, it would strengthen the conclusion that we have measured C-14 concentrations significantly above background in these samples. However, as we discuss below in section I.2, this conclusion is already true at very high confidence when the aggregate approach is used. Thus, the choice of a close-in-time vs. aggregate blank correction approach does not have a significant effect on the main conclusion of the work.

Taking these considerations into account, we concluded that blank correction using a lognormal distribution derived from the aggregate approach was more consistent with (i) the analysis of time-dependent vs. non-time-dependent variability showing that non-time-dependent variability is more important, (ii) suggestions in previous studies that C-14 measurement background is long-tailed, and (iii) the importance of minimizing assumptions about the unknown physical processes leading to blank variability. In addition, as we discuss later in this response, we found that the blank distribution inferred from an aggregate approach was consistent with scatter in replicate measurements, whereas a narrower distribution that would be inferred from a close-in-time approach would not be consistent. Finally, we noted that the main risk of the aggregate approach is a false non-detection of C-14 in our subglacial bedrock cores, so from this perspective the use of the aggregate approach can be considered a stricter or more conservative test of the hypothesis that there was a middle Holocene ice thickness minimum. Thus, we proceeded with the aggregate approach.

**I.2. Detection limits**

The issue of detection limits is complicated in this study because the concentration of cosmogenic C-14 must decrease with depth in each core. Thus, depending on the exposure time and core length, one might commonly expect to encounter a situation where the C-14 concentration in a core top sample was above some defined detection limit at high confidence, but the C-14 concentration in a core bottom sample was not. The question of whether C-14 is present in a core is different from the question of whether it is above some detection limit in one sample from that core. Thus, in this section we will approach the question of whether measured C-14 concentrations are distinct from background by looking at the entire data set at once, rather than individual samples.

[Figure]

*Figure 3. Aggregate distribution of total C-14 atoms observed in all measurement blanks (top panel) compared with distribution of total C-14 atoms observed in all core samples (bottom panel).*

First, Figure 3 compares the aggregate distribution of the total amount of C-14 observed in the 2-year series of blanks with the distribution of the total amount of C-14 (total atoms, not atoms/gram) measured in quartz from core samples. Because the core samples are not all expected to have the same amount of C-14, the core sample distribution is not expected to have any particular form. However, it is evident that these are distinct distributions. Using only the observed distribution of blank concentrations and making no assumptions about the form of the distribution, 91% of the sample measurements are above blank at 68% confidence (see Fig. 3), or, 59% of the sample measurements are above blank at 95% confidence. The probability that one could obtain this result by randomly sampling the blank distribution is effectively zero. Therefore, C-14 is present in the core samples in excess of measurement background.

[Figure]

*Figure 4. Aggregate distribution of total C-14 atoms observed in all measurement blanks compared with distribution of total C-14 atoms observed in core samples from H1 and H4. These data are the same as in Figure 3, but this presentation highlights that samples from near the top of core H1 are well above background.*

Figure 4 shows the same data that are in Figure 3, except by core and in stratigraphic order, which highlights the issue of how to separate the detection limit for C-14 in a sample considered by itself from the detection limit for C-14 in a core. Consider core 19-KP-H1. All replicates of core top samples exceed the 95th percentile of the blank distribution, but some replicates of samples from the bottom of the core are close to the mode of the blank distribution. *However, if cosmogenic C-14 is present in the core top, it*

*is physically required that it be present throughout the core, so even if some replicate analyses would be indistinguishable from blank if considered by themselves, the data from core H1 considered together requires that we have measured C-14 above blank.* This relationship is not as clear for 19-KP-H4 because there is less variation with depth in this core, but it is still the case that 88% of the measurements from H4 exceed the 67th percentile, and 38% exceed the 95th percentile, of the blank measurements, so it is nearly impossible that we could obtain these results if C-14 was not, in fact, present above measurement background. We conclude that C-14 is present above measurement background.

[Figure]

*Figure 5. Total C-14 atoms measured in blanks and core samples during core sample measurement in 2021.*

Finally, Figure 5 shows total C-14 atoms measured in blanks and core samples during the period of core sample measurement in 2021. If, as the review suggests, a close-in-time blank correction were used, the distribution of concentrations observed in the core samples would be higher in relation to the blank distribution used for correction. Thus, a close-in-time blank correction would increase, not decrease, confidence in the conclusion that C-14 concentrations in core samples are above measurement background.

**I.3. Replicate scatter**

The final point in the section of the review dealing with C-14 blank handling has to do with the scatter in replicate measurements. For the core samples, scatter among replicate measurements is extremely large in relative terms, exceeding a factor of 2 for some samples. This observation, taken out of context by itself, would tend to lower confidence in the results. However, we investigated this and found that the replicate scatter in the core samples is exactly as we expect from the measured distribution of measurement blanks.

One way to quantify scatter in the core samples is to observe that because all the samples are closely-spaced sections of the same core, their true C-14 concentrations must vary along a smooth curve (which can be reasonably well approximated over a short depth range by an exponential). Thus, all samples from a single core can in effect be considered a large set of replicates, and the expected distribution of a large set of replicate analyses can be inferred from the distribution of the residuals with respect to a best-fitting exponential.

[Figure]

*Figure 6. C-14 "concentrations" from cores H1 and H4, with best-fitting exponentials. We quote "concentrations" because these values are the total number of atoms present divided by the mass analyzed, so have units of concentration, but have not been blank-corrected, so are not the true concentrations in the samples. The purpose of converting to concentration units is to normalize for small variations in sample mass before fitting an exponential curve.*

[Figure]

*Figure 7. C-14 residuals (atoms) with respect to mean for the aggregate distribution of blanks compared with residuals with respect to a best-fitting exponential for uncorrected concentrations for cores H1 and H4. Residuals computed from Figure 6 have units of atoms/g, but have been transformed back to units of atoms for comparison with the blank distribution. This highlights that the scatter is similar for all three data sets.*

Figures 6 and 7 show the results of this exercise. The distributions of the residuals around a smooth curve for both cores H1 and H2 have essentially the same range as the distribution of blanks around their mean. Although there is no reason to believe that any of these distributions are normal in nature (in fact, we propose that a lognormal distribution is most appropriate for the blank correction and use it throughout the paper), one can compare the distributions in a simple way by comparing standard deviations, which are 35000, 32000, and 46000 atoms for the blanks, H1, and H4 respectively. These have similar magnitude (the higher standard deviation for the H4 data is mostly explained by one outlier), which shows that the scatter among replicate analyses of core samples is as expected if scatter in measurement blanks is the dominant contributor.

It is also possible to look at replicate scatter based on replicate analyses of individual core samples alone, without assuming a relation between adjacent core samples. For this experiment we generate a

distribution of replicate differences by compiling all the differences between combinations of replicate analyses of each sample to define a distribution for replicate scatter. We compare this to a synthetic distribution developed from the measured distribution of blanks by choosing a large number of random blank pairs and recording the difference between them.

[Figure]

*Figure 8. Compiled differences between measured replicates compared with synthetic distribution of expected replicate differences generated by random draws from the distribution of blanks.*

Figure 8 shows this experiment. Although the distribution of observed replicate differences uses fewer data than in Figs. 6-7 so is sparser, the conclusion is the same in that the range of the two distributions shown in Fig. 8 is similar. If we approximate these as normal distributions, they would have standard deviations of 46000 and 48000 atoms, respectively, which are the same. Again, the conclusion is that the scatter in the replicate analyses of core samples is exactly as expected if scatter in the measurement blanks is the dominant contributor.

We conclude that replicate scatter in the core samples is indistinguishable from scatter in the aggregate distribution of measurement blanks. *This is exactly as expected, because at low concentrations, measurement uncertainty should be dominated by the uncertainty in the measurement blank.*

This last point brings up another issue mentioned in the review, which is the apparent scatter observed in C-14 concentrations from erratic and bedrock samples above the present ice surface. By "scatter" in this context, we mean the fact that these results do not lie along a smooth age/elevation curve. It is important to note that, in contrast to the sets of samples from bedrock cores, these samples are not physically required to lie along a smooth curve in age-elevation space. This would only be the case given several fairly restrictive assumptions, including (i) a monotonic and smooth ice thinning history, (ii) a common thinning rate of the last several meters of ice overlying each sample during thinning, and (iii) zero shielding or cover by ice, snow, sediment, or other rocks following initial exposure by ice retreat. There is no reason to believe a priori that these assumptions are strictly true. For these data, as pointed out in the review and also in our manuscript, and as is evident by inspection of the uncertainty bars in Figs. 4 and 7 (Figs. 4 and 7 in the paper, not in this response), the scatter around a smooth age-elevation curve is about twice as large as can be accounted for by scatter in measurement blanks. Although there are other sources of measurement error that could potentially contribute (see discussion of CRONUS-A below), we conclude that the samples do not, in fact, lie along a smooth age-elevation curve. As there is no physical expectation that they should, this is not concerning or surprising, and is not evidence that our blank

correction scheme is correct or incorrect. As described in the paper, we attribute this variation around a smooth age-elevation curve to unsteady ice thinning and variations in the size and shape of ice-marginal snow- and icefields during deglaciation.

To summarize this section, we believe we have thoroughly justified our approach to blank handling with the reasoning above. As we suggest above, we think the best place for this highly technical discussion is in the online discussion, not the text of the paper. Thus, we have not proposed any revisions to the paper here. As noted, we have amended the supplementary data so that all the code and data needed to replicate the analysis and figures above is included in MATLAB workbooks.

**II. CRONUS-A.**

The answer to the main question in this part of the review is yes, the production rate calibration in this study is unchanged from that in previous papers, including among others Goehring et al. (2019) as mentioned in the review. Although additional CRONUS-A measurements have been made at Tulane since that time, they are not significantly different than the data previously used for production rate calibration.

[Figure]

*Figure 9. CRONUS-A measurements during the period in which measurements for this study were made. The x-axis in this figure is the same as in Figure 1. The horizontal line with error bounds is the mean and standard deviation of the set of Tulane CRONUS-A measurements made prior to 2019 and used for production rate calibration in this and previous work. Error bars reflecting AMS and blank measurement uncertainty are not visible at this scale.*

Figure 9 shows these data. As with all other measurements we have discussed here, scatter in excess of nominal analytical uncertainty is present, and because this sample has a much higher C-14 concentration than the samples measured in this study, this scatter cannot be explained by the distribution of process blanks and is therefore unexplained. However, when these measurements are considered together there is no significant difference from the value that was used for production rate calibration in previous work and in this paper. We did not update the production rate calibration for two reasons. First, on general principles, maintaining continuity in production rate estimates to the extent possible makes it easier for readers, so there is no strong reason to continually update the production rate with new data if the changes implied are not significant. Second, the production rate calibration is minimally to negligibly important in this paper. The reason for that is that the paper focuses on samples from subsurface cores having low C-14 concentrations, so the uncertainty in these measurements is dominated by the uncertainty in the blank subtraction. As discussed above, the uncertainty derived from blank subtraction and expressed as reproducibility of replicates and samples from the same core can be up to a factor of 2. However, the entire range of CRONUS-A concentrations observed in multiple laboratories is about 15%, and the total range in CRONUS-A results from Tulane across several years is about 10%. Furthermore, tf

the production rate used in our calculations varied by 10%, it could result in no more than a 10% difference in the duration of the ice thickness lowstand inferred from model fitting. As the length of the lowstand is only constrained by the model fitting to be in the range of 3000-8000 years, an additional 10% uncertainty is insignificant.

Again, we agree that this is important contextual information for the performance of the Tulane lab in a general sense, but as we have discussed above, this issue is not relevant to any of the conclusions in the paper. Thus, we think this discussion is also sufficient without changes to the main text of the paper. We will amend the supplementary data to include the CRONUS-A measurements made in 2019-20.

**III. Comments on model fitting.**

In this section of the review, Lifton asks for our opinion on why there appear to be some systematic misfits between model predictions and Be-10 data in the lower portions of cores H4 and H5. Basically, models that fit the entire data set well appear to underpredict Be-10 data in these areas. Although the well-fitting model predictions are not significantly outside the measurement uncertainty for most of the samples considered individually, predictions are systematically below measurements for several samples in the bottom of both cores. We have, in fact, thought about this, and we see two possible reasons. One possible reason is that, for some reason, the model is predicting that ice is too thin during the lowstand period. This could arise from inaccurate estimates of C-14 production by muons. If this were the case, the effective attenuation length of predicted Be-10 concentrations would be too short. However, this is not a great explanation because if this were true, it would imply that the lowstand was longer than predicted by well-fitting models: if we increase the ice thickness to reconcile predicted and observed attenuation lengths, we have to also increase the duration of the lowstand so that we can continue to match the observed concentrations. As the lowstand in the well-fitting models already takes up all the available time permitted by the constraints at each end of the period during which a lowstand could have taken place, it is not actually possible to increase the lowstand duration and the ice thickness to better fit the Be-10 data. Thus, this explanation fails.

A more likely explanation is that our assumption that the model should start with a zero Be-10 concentration is oversimplified. Even in a situation where subglacial erosion takes place during glacial maximum conditions and removes nearly all Be-10 associated with any hypothetical surface exposure in Pleistocene interglacials (as we believe is the case here), the predicted Be-10 concentration is not zero. If we assume that for the past few million years, this site has, on average, been covered by several hundred meters of ice, and also is subglacially eroded by a few meters during every glaciation, millions of years of exposure at a depth of hundreds of meters would still be expected to lead to hundreds to thousands of atoms per gram of Be-10 in production-decay equilibrium with subsurface production by fast muons. For example, Stone et al. (2019) observed 3500 atoms/g Be-10 in a sample 8 m below the surface of bedrock that is covered by 150 m of ice in present interglacial conditions. It is likely that a nonzero amount of such background subsurface Be-10 is present in our cores and not accounted for in our model.

As the presence of background Be-10 is likely from first principles, but the amount is largely unconstrained, it would be possible to add a background Be-10 concentration that was constant throughout the core depth as an additional free parameter in our model. If produced at hundreds of meters depth, it would be essentially constant over a 1.5-m depth range. We experimented with this, and found that a background Be-10 concentration near 1000 atoms/g slightly improved the fit to Be-10 data in cores H4 and H5. For example, for core H4, the best attainable value of M is about 15% lower if a background Be-10 concentration of 1000 atoms/g is included. This experiment can be made with the MATLAB code in the supplemental data by changing the value in line 66 of 'random_search_wrapper.m'.

However, the improvement in the fit was not large, and there is no significant effect on the best-fitting lowstand duration and ice thickness (without background Be-10, models that fit H4 best have lowstands 5000-8000 years long with 5-8 m of ice, whereas with background Be-10, best-fitting models have 6000-8000 years with 6-9 m of ice). Thus, we took this parameter out of our model on the grounds that it added additional complexity but did not change the results or increase the explanatory power of the model. However, we believe that unaccounted-for background Be-10 is probably the most likely explanation for the small misfit.

As in the previous sections of this review, we think this discussion is important for readers who might wish to duplicate the model-fitting calculations using our MATLAB code, but not highly relevant for most readers of the paper. Thus, we again propose that the public online discussion is the appropriate place for this, and propose no revisions to the main text.

**4. Minor comments.**

This  review made 4 minor corrections and suggestions (the 5th replicates material covered above), which we will correct in the revised version.

---

## Author Response (AR1)

**Authors' response on 'Reversible ice sheet thinning in the Amundsen Sea embayment during the late Holocene', by Balco, Brown, Nichols, Venturelli et al.**
* * *
Dec 19, 2022

Here we index the revisions proposed in the authors' responses to locations in the revised MS. The line numbers refer to the revised manuscript, *not* the latexdiff comparison of original and revised versions.

**Briner review:**

**1**. Clarify reasoning about irreversibility/grounding line position vs. thinning in Section 1 and elsewhere

- Revised text and broke out a new paragraph at page 2, lines 5-22.

**2**. Explain clearly about cold-based ice/no erosion during late Holocene cover in Sections 1 and 2

- Noted that the ice-bed interface is frozen at p. 9, line 8+
- Added a paragraph with a detailed discussion of subglacial erosion at p. 14, line 20

**3**. Additional context for 'irreversible' in Section 1

- Included in revision at p. 2, lines 5-22

**4**. Clarify cosmogenic-nuclide vs. OSL sensitivity in Section 1

- Revised text at p. 2, lines 22-31

**5**. Clarify that ice velocity representation is nonquantitative in Fig 1 caption

- Edited Fig. 1 caption

**6**. Clarify that profile axes are truncated in Fig 2 caption

- Edited Fig. 2 caption

**7**. Remove 'stable' on p. 6

- Corrected at p. 4, line 33

**8**. Provide more context for loose rock recovered in fourth borehole

- On further investigation, our response to this review comment was inaccurate. We went back and looked at notes and photos and in fact we could not verify that basal ice was present below the rock fragments that were recovered. We now make this clear at p. 10, lines 4-6.

**9**. Clarify detection limit issue for C-14

- This is now clarified at p. 12, lines 25-26. In addition, we discussed this issue at length in the response to the Lifton review.

**10**. Clarifications and editing for concluding remarks in Section 6

- We thoroughly revised the concluding section on p. 21.

Finally, this review requested photos of the rock cores. Although, as we indicate in the response, these are not actually very interesting – they just look like rocks that could be from anywhere – and they are already available online, it would be possible to include core photos in the supplementary data. The supplement is currently 7.1 MB, and the core photos would add an additional 60 MB. Please let us know if you would like us to add these photos.

**Lifton review:**

As we discussed by email, (i) we included a thorough response to all of the questions and comments in this review in the authors' response, but (ii) as all the material in this response is highly technical and relevant only to certain aspects of the C-14 measurements, we have not included it in the main text of the paper. As we also discussed by email, we included a citation to the review response at p. 12, line 30. Finally, as indicated in the response, we added to the supplement all the MATLAB code needed to generate all figures and results shown in the response, as well as the requested data on Tulane measurements of the CRONUS-A standard. Although we did not revise any of our blank correction calculations as suggested by Lifton, we believe we have thoroughly explained the reasoning behind the approach that we have adopted, and we believe we have addressed all the concerns in his response in detail.

In addition, this review made 4 minor typographical corrections and suggestions (the 5th relates to technical aspects of C-14 measurements as discussed above), and these have all been corrected in the revised MS.

---

## Author Response (AR2)

**Authors' response to reviewers (second round) for TC 2022-172, "Reversible ice sheet thinning in the Amundsen Sea Embayment during the Late Holocene."**
* * *
Mar 15, 2023

This response documents corrections to the paper in response to a second review by Nat Lifton. We appreciate the close attention of this reviewer to these technical items.

**NOTE: We have not supplied a complete latexdiff comparison of previous and revised versions of the main text, because there are only three minor technical corrections in the main text.**

There are additional corrections to two of the supplementary files, which are also listed below.

**Corrections to main text:**

(1) Reviewer comment: *On page 12, line 30, author comment AC1 erroneously cited instead of AC2.*

– Corrected in reference list on p. 23, line 17.

(2) Reviewer comment: *"P. 12, Line 15: The currently accepted half-life for 14C is 5700 ± 30 yr (www.nndc.bnl.gov) – see Hippe and Lifton (2014) – this was corrected on page 4 but not here."*

– Corrected '5730' to '5700' on p. 12, line 15.

(3) Reviewer comment: *"P. 12, Line 20: The 14C is not extracted in vacuum but rather in an atmosphere of ca. 50 torr (66 mbar) of high-purity O2. The procedure is correct in Supplementary Methods 2 - this should reflect that."*

– Corrected 'in vacuum' to 'in a high-purity $O_2$ atmosphere" on p. 12, line 20.

**Corrections to Supplementary Table S5:**

(4) Reviewer comment: *"However, they only present total atoms measured for each analysis, when the measured concentration is the key value for intercomparison. Also, when I calculate the concentrations of CRONUS-A from the quartz mass and total 14C atoms, for the previously published samples the values differ significantly from those presented in Table 4 of Goehring et al. (2019). An explanation should be provided if calculations were done differently for this table. "*

– Added notes to Table S5 clarifying how to reconcile the data in this table with those reported in Goehring (2019).

(5) Reviewer comment: *"In addition, for ID CA040417, the supplemental spreadsheet notes an anomalously high CO2 yield, but Goehring et al. (2019) indicate in their Table 4 that the sample was diluted prior to extraction with synthetic graphite. The note should be consistent with that explanation if that is the case. "*

– Added notes on this issue to Table S5.

(6) Reviewer comment: *"Supplement S5.xls: Indications of thick or thin tube on each worksheet should be explained in the notes on those sheets (or at least the first one of them). "*

– Added appropriate notes to Table S5.

**Corrections to Supplementary Methods 2:**

(7) Reviewer comment: *"In my original review, I also thought it would be important that the authors include a discussion of the CRONUS-A 14C measurements relative to nominal values of Jull et al. (2015), as they do for both 10Be and 26Al in Supplemental Methods 2 document. While they are citing the discussion on this topic in the Authors' Response to Reviewer 2 (me) dated Nov 22, 2022 in the main paper, they did not include any summary discussion paragraph in the 14C section of Supplementary Methods 2, which I thought would be an appropriate place. At the very least I think they should cite the AC2 discussion in the Supplementary Methods 2 document in the section discussing in situ 14C and CRONUS-A, to point the interested reader in that direction. "*

– Added a citation to the Author Comment in the section of Supplementary Methods 2 headed "blank correction."